# $k$-Means Clustering with Distance-Based Privacy

**Alessandro Epasto**
Google Research
aepasto@google.com

**Vahab Mirrokni**
Google Research
mirrokni@google.com

**Shyam Narayanan**
MIT
shyamsn@mit.edu

**Peilin Zhong**
Google Research
peilinz@google.com

## Abstract

In this paper, we initiate the study of Euclidean clustering with Distance-based privacy. Distance-based privacy is motivated by the fact that it is often only needed to protect the privacy of exact, rather than approximate, locations. We provide constant-approximate algorithms for $k$-means and $k$-median clustering, with additive error depending only on the attacker's precision bound $\rho$, rather than the radius $\Lambda$ of the space. In addition, we empirically demonstrate that our algorithm performs significantly better than previous differentially private clustering algorithms, as well as naive distance-based private clustering baselines.

## 1 Introduction

Two of the most fundamental and widely studied problems in unsupervised machine learning are the $k$-means and $k$-median clustering problems. Solving these clustering problems can allow us to group together data efficiently, and hence extract valuable and concise information from massive datasets. The goal of the $k$-means (resp., $k$-median) clustering problem is: given a dataset $X$ of points, construct a set $C$ of $k$ centers to minimize the clustering cost $\sum_{x \in X} d(x, C)^2$ (resp., $\sum_{x \in X} d(x, C)$), where $d(x, C)$ represents the minimum distance between the data point $x$ and the closest center in $C$.

In general, machine learning and data mining algorithms are prone to leaking sensitive information about individuals who contribute data points. In certain scenarios, this can lead to severe consequences, including losses of billions of dollars [60] or even the loss of human lives [10]. Thus, providing accurate algorithms that protect data privacy has become crucial in algorithm design. Over the past decade, the notion of differential privacy (DP) [31] has emerged as the gold standard for privacy-preserving algorithms, both in theory and in practice, and has been implemented by several major companies and the US Census [34, 68, 30, 1]. Informally, DP requires the output distribution of the algorithm to remain almost identical whenever a single data point is altered. (See Section 2 for a formal definition.) Hence, even the knowledge of all but one data point, along with the output of the algorithm, still cannot reveal significant information about the final data point.

The importance of $k$-means and $k$-median clustering, as well as preserving data privacy, has led to a large interest in designing differentially private clustering algorithms in Euclidean space [14, 62, 36, 47, 59, 73, 64, 65, 71, 38, 9, 63, 48, 70, 69, 44, 49, 17, 61, 19, 13, 24, 33, 25, 56]. Here, the goal is to design a differentially private set of $k$ centers, such that the clustering cost with respect to these centers is only a small factor larger than the optimal (non-private) clustering cost. Importantly, the work of [70, 44, 25] led to efficient polynomial-time and differentially private algorithms that achieve constant multiplicative approximation ratios.

While we can obtain DP algorithms with low multiplicative error, all such algorithms also require an additional additive error. If $\Lambda$ is the radius of a ball that is promised to contain all data points, even the

best private clustering algorithms are known to have an additive error proportional to $\mathrm{poly}(k, d) \cdot \Lambda^p$, where $p = 2$ for $k$-means and $p = 1$ for $k$-median. This factor of $\Lambda^p$ is in fact unavoidable [47], as a single individual datapoint can be moved up to distance $\Lambda$ and the algorithm must preserve privacy with respect to this change. If we do not have a good bound of $\Lambda$, this factor may dominate the error, and may make the clustering algorithm highly inaccurate. Even if the bound is known exactly, errors scaling with $\Lambda$ may however be unnecessary and unacceptable in certain situations.

The additive error depending on $\Lambda$ is necessary because standard differential privacy requires us to protect learning *anything* about the location of any point. However, in practice this may not be necessary as it might be enough to not know the location of a point up to a certain error. For instance, in address data, the risk is leaking the actual location, but uncertainty within a few miles in a city is sufficient to protect the privacy of the person [20]. Another motivation is in smart meters [20, Section 6.1], where accurately learning the fine-grained consumption can result in spectacular privacy leaks (e.g. learning which TV channel is being watched [46, 52]) but slight uncertainty on the measurements is sufficient to protect from such attacks. Moreover, when differential privacy is used to protect the algorithm from adversarial inputs, it is often sufficient to protect against small perturbations as large perturbations can be detected or removed otherwise [53].

These cases can be modeled by variants of differential privacy, such as dX privacy (a.k.a. extended differential privacy) [20, 40], and pixelDP [53]. All such models are adaptations or generalizations of DP which take into account a metric over the datasets.

In this paper, we study a concrete formulation of distance-based privacy which we call $\rho$-dist-DP. Roughly speaking, an algorithm is $\rho$-dist-DP if the algorithm protects privacy of a single data point if it is moved by at most $\rho$ in a metric space. (See Section 2 for a formal definition, where we define $(\varepsilon, \delta, \rho)$-dist-DP.) This is a less restrictive version of DP, as usually the neighboring datasets are defined to be any two datasets with a single point allowed to move anywhere. While we remark that although this notion is well-defined for any metric space, our results in this paper focus entirely on Euclidean space.

The main question we study in this paper is the following: can we obtain much better approximation results (and algorithms better in practice) if we allow the algorithm to resist small movements, as opposed to arbitrary movements, of a point for instance for clustering? In other words, can we design $\rho$-dist-DP algorithms that perform significantly better than the state of the art regular DP algorithms for $k$-means or $k$-median clustering?

## 1.1 Our Results

In this work, we answer the above question affirmatively, by providing an efficient and accurate theoretical algorithm, and showing empirically that our algorithm outperforms clustering algorithms with standard differential privacy.

### 1.1.1 Theoretical Results

From a theoretical perspective, we are able to obtain $O(1)$-approximate algorithms for $k$-means and $k$-median clustering with $\rho$-dist-DP, and with additive error essentially only depending on the smaller distance $\rho$ as opposed to the full radius $\Lambda$. More precisely, our main theorem is the following.

**Theorem 1.1.** *Let $n, k, d$ be integers, $\rho \in (0, \Lambda], \varepsilon, \delta \in (0, 1]$ be privacy parameters, and $p \in \{1, 2\}$. Then, given a dataset $X = \{x_1, \ldots, x_n\}$ of points in a given ball of radius $\Lambda$ in Euclidean space $\mathbb{R}^d$, there exists a polynomial-time $(\varepsilon, \delta, \rho)$-dist-DP algorithm $\mathcal{A}$ that outputs a set of centers $C = \{c_1, \ldots, c_k\}$, such that*

$$\sum_{i=1}^{n} d(x_i, C)^p \leq O(1) \cdot \min_{\substack{C^* \subset \mathbb{R}^d \\ |C^*| = k}} \sum_{i=1}^{n} d(x_i, C^*)^p + \mathrm{poly}\left(k, d, \log n, \frac{1}{\varepsilon}, \log \frac{1}{\delta}, \log \frac{\Lambda}{\rho}\right) \cdot \rho^p.$$

*Here, $p = 1$ for $k$-median and $p = 2$ for $k$-means.*

For more precise dependences on the parameters $k, d, 1/\varepsilon$, please see Theorem C.1.

Qualitatively, Theorem 1.1 has similar guarantees to [70], who also provided an $(\varepsilon, \delta)$-differentially private algorithm with an $O(1)$-approximation algorithm and additive error that was

$\text{poly}(k, d, \log n, \frac{1}{\varepsilon}, \log \frac{1}{\delta}) \cdot \Lambda^p$. The main difference is that we drastically reduce the additive error by reducing the dependence on $\Lambda$ to a dependence on the distance privacy parameter $\rho$.

**Running time and parallel computation.** The runtime of a straightforward implementation of our algorithm is $\tilde{O}(nkd) + \text{poly}(k) \cdot d$,[1] if we also ignore polynomial factors in $\log \frac{\Lambda}{\rho}$. By using approximate near neighbor algorithms, we can improve this further to $\tilde{O}(nd) + \text{poly}(k) \cdot d$, which for $k$ at most a small polynomial in $n$, is nearly linear. In addition, the algorithm can be easily implemented in the massively parallel computation (MPC) model [51, 11] (an abstraction of MapReduce [28]) using $O(1)$ rounds and near linear total space where each machine has sublinear space. We discuss above in-memory algorithms and MPC algorithms further at the end of Appendix C.

Finally we remark that the $\rho^p$ dependence in the additive error is required for ensuring $\rho$-dist-DP. In fact, we prove in Appendix D that any $(\varepsilon, \delta, \rho)$-dist-DP algorithm, with any finite multiplicative error, must incur $\Omega(k \cdot \rho^2)$-additive error for $k$-means and $\Omega(k \cdot \rho)$-additive error for $k$-median.

### 1.1.2 Empirical Results

We empirically studied the performance of our algorithm on public and real-world datasets. We compare the approximation guarantee of our algorithm with the standard DP clustering algorithm and the standard non-private $k$-clustering algorithm. Experiments show that our algorithm outperforms the DP clustering algorithm and is only slightly worse than the non-private algorithm. In addition, we show that smaller $\rho$ provides a better approximation guarantee, which aligns with our theoretical study. We refer readers for more details of our empirical study to Section 6.

### 1.2 Other Related Work

**Distance-based Privacy:** The literature on distance-based privacy explored different data protection schemes which we now describe in more detail. A general notion is known as dX privacy [20] (a.k.a. Extended differential privacy) which includes as a special case differential privacy. This privacy notion bounds the distinguishability of two statistical datasets, not just by the number of different users' inputs (i.e., their Hamming distance) but by an arbitrary $d_\chi$ distance between them accounting for the magnitude of the changes to each user entry. Similar notions, such as pixelDP [53] and perceptual indistinguishability [21], are also formalization of DP where adjacent datasets differ in a single feature of the input (e.g. a pixel) or some custom function of the data. Several algorithms have been defined for these notions, including LSH algorithms [40].

From an application point of view, much work has focused on geo-indistinguishability [3, 6, 15], i.e. preventing an adversary from distinguishing two close locations (by ensuring that close location have similar probabilities to generate a certain output). Other areas of applicability has been protecting textual data [39, 41], private smart meters sensing [27], image obfuscation [35, 21] and mobile crowsensing [74].

**$k$-Clustering:** $k$-Means and $k$-median clustering have seen a large body of work over the past few decades. While both problems are known to be NP-hard [58], a significant amount of work has given various $O(1)$-approximation algorithms for both problems [18, 8, 50, 7, 55, 16, 2, 26]. The state-of-the-art approximation is a $5.912$-approximation for Euclidean $k$-means and a $2.406$-approximation for Euclidean $k$-median [26]. As noted in previously, there has also been significant work in specifically studying *differentially private* $k$-means and $k$-median clustering, though to our knowledge we are the first to study distance-based private clustering.

## 2 Preliminaries

We present some basic definitions and setup that will be sufficient for explaining our algorithms for the main body of the paper. We defer some additional preliminaries to Appendix A.

---

[1] $\tilde{O}(f(n))$ denotes $O(f(n) \log f(n))$.

## 2.1 Differential Privacy

First, we recall the definition of differential privacy.

**Definition 2.1.** [31] A (randomized) algorithm $\mathcal{A}$ is said to be $(\varepsilon, \delta)$-*differentially private* $((\varepsilon, \delta)$-DP for short) if for any two datasets $X$ and $X'$ that differ in exactly one data point and any subset $S$ of the output space of $\mathcal{A}$, we have

$$\mathbb{P}(\mathcal{A}(X) \in S) \le e^\varepsilon \cdot \mathbb{P}(\mathcal{A}(X') \in S) + \delta.$$

In standard differential privacy, two datasets $X$ and $X'$ are *adjacent* if we can convert $X$ to $X'$ either by adding, removing, or changing a single data point. Notably, the change in the single data point may be arbitrary.

In distance-based privacy, however, we only allow two datasets to be adjacent if they differ by changing (not adding or removing) a single data point, by moving it up to distance $\rho$. Formally, we define the following.

**Definition 2.2.** Let $X, X'$ be $\rho$-*adjacent* if they have the same number of points and differ in exactly one data point, where the distance between the two differing data points is $\rho$. Then, a (randomized) algorithm $\mathcal{A}$ is $(\varepsilon, \delta, \rho)$-dist-DP if for any two $\rho$-adjacent datasets $X$ and $X'$ and any subset $S$ of the output space of $\mathcal{A}$, we have

$$\mathbb{P}(\mathcal{A}(X) \in S) \le e^\varepsilon \cdot \mathbb{P}(\mathcal{A}(X') \in S) + \delta.$$

We remark that in all of our theoretical guarantees, we implicitly assume that $\varepsilon, \delta \le \frac{1}{2}$.

The Laplace Mechanism is one of the most common primitives used to ensure privacy. Simply put, for a non-private statistic, the Laplace Mechanism adds noise $\mathrm{Lap}(t)$ to the statistic for some $t > 0$, where $\mathrm{Lap}(t)$ has the probability density function (PDF) equal to $\frac{1}{2t} \cdot e^{-|x|/t}$. It is well-known that if $f(X)$ is a statistic such that $|f(X) - f(X')| \le \Delta$ for any two adjacent datasets $X, X'$, then $f(X) + \mathrm{Lap}(\Delta/\varepsilon)$ is $(\varepsilon, 0)$-DP. Likewise, if $|f(X) - f(X')| \le \Delta$ between two $\rho$-adjacent datasets $X, X'$, then $f(X) + \mathrm{Lap}(\Delta/\varepsilon)$ is $(\varepsilon, 0, \rho)$-dist-DP.

Similar to the Laplace Mechanism, we can also implement the *Truncated Laplace mechanism* [43] for approximating functions $f : X \to \mathbb{R}$. The Truncated Laplace Mechanism outputs $f(X) + \mathrm{TLap}(\Delta, \varepsilon, \delta)$, where $\mathrm{TLap}(\Delta, \varepsilon, \delta)$ is the distribution with PDF proportional to $e^{-|x| \cdot \varepsilon/\Delta}$ on the region $[-A, A]$, where $A = \frac{\Delta}{\varepsilon} \cdot \log\left(1 + \frac{e^\varepsilon - 1}{2\delta}\right)$, and PDF 0 outside the region $[-A, A]$. Assuming $0 < \varepsilon$ and $0 < \delta \le \frac{1}{2}$, it is known that if $|f(X) - f(X')| \le \Delta$ for all adjacent $X, X'$, then this mechanism is $(\varepsilon, \delta)$-DP, and if $\varepsilon \le \frac{1}{2}$ this is accurate up to error $\frac{\Delta}{\varepsilon} \cdot \log \frac{1}{\delta}$, with probability 1.

Likewise, a nearly identical result holds for distance-based privacy. Namely, if $|f(X) - f(X')| \le \Delta$ for any $\rho$-adjacent datasets $X, X'$, then $f(X) + \mathrm{TLap}(\Delta, \varepsilon, \delta)$ is $(\varepsilon, \delta, \rho)$-dist-DP.

We defer some additional preliminaries to Appendix A.

## 2.2 $k$-Means and $k$-Median Clustering

We define $d(x, y)$ to be the Euclidean distance between two points $x$ and $y$, and for a finite subset $C \subset \mathbb{R}^d$, we define $d(x, C) = d(C, x)$ to be $\min_{c \in C} d(x, c)$. Given a dataset $X = \{x_1, \ldots, x_n\}$ of points in $\mathbb{R}^d$, and a set of centers $C = \{c_1, \ldots, c_k\}$, we define the $k$-means/$k$-median cost as

$$\mathrm{cost}(X; C) := \sum_{x \in X} d(x, C)^p.$$

Above, $p = 2$ for $k$-means and $p = 1$ for $k$-median. Finally, we define $\mathrm{OPT}_k(X)$ to be the minimum value of $\mathrm{cost}(X; C)$ for any set of $k$ points $C$.

We further assume that the points in $X$ are in $B(0, \Lambda)$, which is the ball of radius $\Lambda$ about the origin in $\mathbb{R}^d$. Our goal in $k$-means (resp., $k$-median) clustering is to find a subset $C$ of $k$-points that minimizes $\mathrm{cost}(X; C)$, i.e., where $\mathrm{cost}(X; C)$ is as close to $\mathrm{OPT}_k(X)$ as possible. Occasionally, we may assign each point $x_i \in X$ a positive weight $w_i$, in which case we define $\mathrm{cost}(X; C) := \sum_{x_i \in X} w_i \cdot d(x_i, C)^p$.

Our goal in differentially private clustering is to produce a set of $k$ centers $C$ such that $C$ is $(\varepsilon, \delta)$-DP with respect to $X$, and such that $\text{cost}(X; C) \leq \beta \cdot \text{OPT}(X) + V \cdot \Lambda^p$ (where $p = 2$ for $k$-means and $p = 1$ for $k$-median), where $\beta$ and $V$ are not too large. In distance-based privacy, we wish to replace the factor $\Lambda$ with some smaller $\rho$, i.e., we want $\text{cost}(X; C) \leq \beta \cdot \text{OPT}(X) + V \cdot \rho^p$. However, our algorithm only has to be private up to changing a single data point by up to $\rho$. If we obtain this guarantee, we say that we have a $(\beta, V)$-approximate and $(\varepsilon, \delta, \rho)$-dist-DP solution.

## 3 Technical Overview and Roadmap

We focus on proving Theorem 1.1 in Sections 4 and 5, and discuss our experimental results in Section 6. In Sections 4 and 5, we will only describe the algorithms, and we defer all formal proofs to the Supplementary sections. For simplicity, in this overview we focus on $k$-median and assume the dimension $d = (\log n)^{O(1)}$, and can be hidden in $\tilde{O}$ notation.

Our approach follows two high-level steps, inspired by the work of [22, 25]. The insight used in [25], which proved highly efficient private clustering algorithms, is to start by generating a crude but private solution that may use a large number of centers and have a large approximation, but has small additive error. Then, one can apply the crude solution to partition the Euclidean space $\mathbb{R}^d$ into smaller regions, and apply some regular differentially private clustering algorithm in the regions. We follow a similar high-level template to [25]. However, we still need to implement each of these steps, which require several technical insights to ensure we maintain privacy while only losing additive error roughly proportional to $\text{poly}(k, d) \cdot \rho$.

To obtain a crude approximation, we use a technique based on partitioning the space $\mathbb{R}^d$ into randomly shifted grids at various levels (also known as the Quadtree). In the Quadtree, the 0th level is a very coarse grid containing the large ball of radius $\Lambda$, and each subsequent level refines the previous level with smaller grid cells. For a single grid and knowledge of which point lies in which grid cell, a natural approach for minimizing cost would be to output the centers of the "heaviest" cells, i.e., those with the most number of points. Indeed, it is known that outputting the $O(k)$ heaviest cells at each grid level provides a good approximation, at the cost of having more than $k$ centers.

While this is not DP, a natural way of ensuring privacy would be to add Laplace noise to each count and add the heaviest cells after this. Unfortunately, doing so will lead to error depending on the full radius $\Lambda$, due to the coarser levels of the quadtree (i.e., levels with grid length close to $\Lambda$ rather than $\rho$). For example, if there was only a single data point, there will be at least $e^d$ cells even at coarse levels, and several of them may have large noisy counts. Hence, we are likely to choose completely random cells, which will cause additive error to behave like $\Lambda$ as opposed to $\rho$. Another option is to add noise to the points first and then compute the heaviest cells. While this avoids additive dependence on $\Lambda$, the additive dependence will behave like $n \cdot \rho$ where $n$ is the full size of the dataset.

Surprisingly, we show that we can *combine* both of these observations in the right way. Namely, for coarse cells (i.e., with length larger than $\tilde{O}(\rho)$), we add noise (of distance proportional to $\tilde{O}(\rho)$) to the data points directly to generate *private points* $\tilde{x}_i$, and then compute the heaviest cells without adding noise to the counts. For fine cells (length smaller than $\tilde{O}(\rho)$), we do not add noise to the data points, but we add Laplace noise to the cell counts.

To explain the intuition behind this, suppose that the $n$ data points happen to be perfectly divided into $n/k$ clusters, where every point has distance $r$ to its nearest cluster center. If $r \gg \rho$, then even if we add $\tilde{O}(\rho)$ noise to each data point, we will still find cluster centers that are within $\tilde{O}(r)$ of each correct center. So, the $k$-means cost should only blow up by a small multiplicative factor, without additive error. Alternatively, if $r \ll \rho$, then the grid cells of side length $\tilde{O}(r)$ should contain the entire cluster, and hence have $n/k$ points in them. Assuming $n \gg d \cdot k$, even if we add Laplace noise to each of $e^d$ cells, none of them will exceed $n/k$. Alternatively, if $n \ll d \cdot k$, then our approach of simply adding noise to the points and obtaining $n \cdot \rho$ error will be only $O(dk) \cdot \rho$, which is small.

In summary, we can generate a crude approximation $F$ with roughly $O(k)$ cells per grid level (and $\tilde{O}(k)$ centers total), with small additive ratio. But we desire for the number of centers to be exactly $k$, and the multiplicative ratio to be $O(1)$, whereas ours will end up being $d^{O(1)}$. To achieve such an accurate result, we use $F$ to partition the data into regions, and apply a private coreset algorithm on

each. By combining these coresets together, we may obtain a private coreset of the full data, and then we can apply an $O(1)$-approximate non-private algorithm on the coreset.

A first attempt, inspired by [22, 25], is to send each $x_i$ to a region $S_j$ if $f_j \in F$ is the closest center to $x_i$, and then compute a standard (i.e., not dist-DP) private coreset on each region $S_j$. To avoid dealing with large additive errors depending on $\Lambda$, we further split each region into a close and far region, depending on whether the distance from $x_i$ to $f_j$ is more than or less than $S \cdot \rho$ for some parameter $S$.

This attempt will still suffer from a large additive cost. For instance, if a point moves, even by distance $\rho$, it may move from a close region to a far region. Hence, the far region may have 1 more point, and since the far regions have diameter $\Lambda$, an algorithm that is private to adding or deleting a point must incur error proportional to $\Lambda$.

Our fix for this is to assign each $x_i$ to a region not based on its closest point and distance, but instead based on $\tilde{x}_i$'s closest point and distance, where we recall that $\tilde{x}_i$ is the noisy version of $x_i$. For the points $\{x_i\}$ that are mapped to a far region (meaning $\tilde{x}_i$ is far from its nearest $f_j$), we will simply use $\{\tilde{x}_i\}$ as the coreset, as $\tilde{x}_i$ is already dist-DP. However, for points that are mapped to a close region, while we use $\tilde{x}_i$ to determine which region the point $x_i$ is mapped to, we compute a private coreset using [70] on the points $x_i$, rather than use the points $\tilde{x}_i$.

To explain why this algorithm is accurate, for the close regions, we obtain additive error proportional to $S \cdot \rho$ as we apply the private coreset on a ball of radius $S \cdot \rho$. There is one region for each center in $F$, which multiplies the additive error by $|F| = \tilde{O}(k)$. For the far regions, we first note that $d(\tilde{x}_i, C) = d(x_i, C) \pm \tilde{O}(\rho)$ for any set of $k$ centers $C$, as $d(x_i, \tilde{x}_i) \leq \tilde{O}(\rho)$. Hence, we have additive error $\tilde{O}(\rho)$ per point. While this seems bad as this might induce additive error for $n$ points, we in fact show that this additive error can be "charged" to multiplicative error. To see why, if $x_i$ mapped to the far regions, this means $d(\tilde{x}_i, F) \geq \rho \cdot S$, which also means $d(x_i, F) \geq \Omega(\rho \cdot S)$, If there were $T$ such points, then the total cost of $X$ with respect to $F$ is at least $T \cdot \rho \cdot S$, whereas the additive error is roughly $T \cdot \rho$. Finally, in our crude approximation we show $\text{cost}(X; F)$ is at most $d^{O(1)}$ times the optimum $k$-means cost, which means for $S \gg d^{O(1)}$ the additive error is small even compared to the optimum cost. Hence, we can charge the additive error to multiplicative error. We still have additive error from the close regions, but for $S = d^{O(1)}$, the additive error is only $\text{poly}(k, d) \cdot \rho$.

To summarize, while our techniques are inspired by [25], one important novel technical contribution of our work is that while [25] uses the true locations of the points to assign them to regions, we first add Gaussian noise to the points to determine their region, and then use the noised points *only* for the "far" regions and the true points *only* for the "close" regions. This change is crucial in ensuring the analysis is successful. In addition, we must set several parameters carefully to charge the additional incurred cost either to a small additive or small multiplicative factor.

## 4 Crude Approximation

In this section, we devise a crude bicriteria approximation that will serve as a starting point in developing our more refined algorithm. A *bicriteria* approximation is a set $F$ of $\alpha \cdot k$ points, that is $(\varepsilon, \delta, \rho)$-DP in terms on $X$, and in addition, it is a $(\beta, V)$ approximation, i.e., $\text{cost}(X; F) \leq \beta \cdot \text{OPT}_k(X) + V \cdot \rho^p$, where $p = 1$ for $k$-median and $p = 2$ for $k$-means. Even though $F$ has more than $k$ points, we still compare to the optimal solution with exactly $k$ points. We will show such an algorithm with $\alpha = \text{poly}(\log n, \log \frac{\Lambda}{\rho})$, $\beta = \text{poly}(d)$, and $V = \text{poly}(k, d, \varepsilon^{-1}, \log \delta^{-1}, \log n)$. We defer the formal theorem statement, along with the proof, to Appendix B.

**Algorithm Description:** The algorithm works as follows. For each $i \leq n$, let $\tilde{x}_i$ be generated by adding $O\left(\frac{\rho}{\varepsilon} \cdot \sqrt{\log(1/\delta)}\right) \cdot \mathcal{N}(0, I)$ noise to each data point $x_i$. Let $\tilde{X} = \{\tilde{x}_1, \ldots, \tilde{x}_n\}$.

We create $REP = O(\log n)$ random quadtrees starting from the top level with side length $\Lambda$ (full diameter of pointset) until the bottom level of size length $\rho/B$, for some parameter $B$. Next, for some parameter $A$, for each level with side length between $\rho \cdot A$ and $\rho/B$, we count how many points are in each cell, add $\text{TLap}(1/\varepsilon', 1/\delta')$ noise, where $\varepsilon' = \Theta(\varepsilon/\sqrt{\log n \log(A \cdot B) \log(1/\delta)})$ and $\delta' = \Theta(\delta/(\log n \log(A \cdot B)))$, and then pick the $4k$ cells in that level with the largest number of points in them, after adding noise to the number of points. For the levels of side length more than $\rho \cdot A$, we count how many of the $\tilde{x}_i$ points are in each cell and then pick the $4k$ cells in that level with

the largest number of points in $\tilde{X}$. Our final algorithm simply outputs the union of all cell centers that we have picked.

One issue is that the number of cells is exponential in $d$, so adding noise to each cell count may be inefficient. To fix this, we will only add $\mathrm{TLap}(1/\varepsilon', 1/\delta')$ noise to cells that were nonempty, and will only pick a cell center if its noisy count is at least $\frac{K}{\varepsilon'} \log \frac{1}{\delta}$, for some large constant $K$. Since an empty cell, even after adding noise to its count, can never exceed $\frac{K}{\varepsilon'} \log \frac{1}{\delta}$, we can pretend we did the same procedure to the empty cells, but simply never included them. It is straightforward to verify that every other step of the algorithm is implementable in polynomial time.

We provide pseudocode for the algorithm in Algorithm 2 in Appendix B, and we discuss the runtime at the end of Appendix C.

## 5   From Crude to Accurate

In this section, we devise an improved approximation that only uses $k$ centers and achieves a constant approximation ratio, using the crude approximation from Section 4 as a starting point. We will subsequently prove Theorem 1.1. Again, we defer all proof details to Appendix C.

Our approach utilizes both the crude approximation from Section 4 and previously known constant-approximation differentially private (but not dist-DP) algorithms from the literature, to create a dist-DP "semi-coreset" for clustering. More formally, given a set of $n$ points $X = \{x_1, \ldots, x_n\} \in \mathbb{R}^d$, we will compute a (weighted) set of points $Y$ that is $(\varepsilon, \delta, \rho)$-dist-DP with respect to $X$, such that for any set of $k$ centers $C = \{c_1, \ldots, c_k\}$, $\mathrm{cost}(Y; C) = \Theta(\mathrm{cost}(X; C)) \pm O(\mathrm{OPT}_k(X)) \pm W \cdot \rho^p$, where $W$ will be polynomial in $d, k, \varepsilon^{-1}, \log \delta^{-1}, \log n$, and $\log \frac{\Delta}{\rho}$.

If we can achieve this, then we just have to compute an $O(1)$-approximate $k$-means (or $k$-median) solution to $Y$, which does not have to be private since $Y$ already is. Indeed, one can prove an $O(1)$-approximation of $Y$ will be a dist-DP $(O(1), O(W))$-approximate solution for $X$.

**Algorithm Description:**   Our algorithm works as follows. First, for each point $x_i \in X$, add $O\left(\frac{\rho}{\varepsilon} \cdot \sqrt{\log(1/\delta)}\right) \cdot \mathcal{N}(0, I)$ noise to get a point $\tilde{x}_i$. (Recall: this was also done for the crude approximation.)

Next, we partition the set of points into regions, using our dist-DP bicriteria approximation $F$ from Section 4. If $d(\tilde{x}_i, F) > \rho \cdot S$ for some parameter $S$, we send the *noised point* $\tilde{x}_i$ to the set $\tilde{X}_0$, and send the index $i$ to the index set $I_0$. Else, if $\tilde{x}_i$ is closest to center $f_j$ (for $j \le \alpha \cdot k$), we send the *true point* $x_i$ to the set $\hat{X}_j$, and send $i$ to the index set $I_j$. In fact, for all $j$ including $j = 0$, we may define $\hat{X}_j$ to be the set $\{x_i : i \in I_j\}$. Note that $\hat{X}_j$ is a full partition of the dataset $X$. For each $j \ge 1$, we will define the region $R_j$ as the ball of radius $O(\rho \cdot S)$ around $f_j$.

For each $0 \le j \le \alpha \cdot k$, we let $\hat{n}_j$ be the number of indices in $I_j$. Note that this equals the number of points mapped to $\hat{X}_j$. If $\hat{n}_j < T$ for some parameter $T$, then we define $\tilde{X}_j$ to be the corresponding points $\{\tilde{x}_i : i \in I_j\}$. Otherwise, we apply the private semi-coreset algorithm from [70] to find a private semi-coreset $\tilde{X}_j$ of the dataset $\hat{X}_j$, with respect to the ball $B(f_j, \rho \cdot O(S/\gamma))$ for some parameter $\gamma < 1$. Finally, we will merge all the semi-coresets $\tilde{X}_j$ together, which includes $\tilde{X}_0$ defined in the previous paragraph, to obtain $\tilde{X}$. Finally, we may apply any $O(1)-$approximate (non-private) clustering to $\tilde{X}$.

We provide pseudocode for the algorithm, in Algorithm 1.

## 6   Empirical Evaluation

In this section, we study the emperical approximation of our $\rho$-dist-DP $k$-means clustering algorithm.

**Datasets.**   We evaluate our algorithm on 6 well-known public datasets *brightkite* ($51406 \times 2$), *gowalla* ($107092 \times 2$), *shuttle* ($58000 \times 10$), *skin* [12] ($245057 \times 4$), *rangequeries* [67] ($200000 \times 6$) and *s-sets* [42] ($5000 \times 2$), where brightkite and gowalla are datasets of geographic locations (latitude and longitude) of users and can be found in Stanford Large Network Dataset Collection (SNAP) [54], shuttle, skin and rangequeries are non-geographic datasets and can be found on UCI Repository [29],

---

**Algorithm 1** Main Algorithm: dist-DP $k$-means (resp., $k$-median) clustering

---

1: **Input:** Parameters $n, d, k, \varepsilon, \delta, \rho$, dataset $X = \{x_1, \ldots, x_n\} \subset \mathbb{R}^d$, crude private bicriteria approximation $F = \{f_1, \ldots, f_{\alpha \cdot k}\} \subset \mathbb{R}^d$.

2: **Output:** Improved private approximation $C = \{c_1, \ldots, c_k\} \subset \mathbb{R}^d$.

3: **Initialize** $S = O\left(\frac{1}{\varepsilon} \cdot \sqrt{(d + \log n) \cdot \log(1/\delta) \cdot d^3}\right), T = O\left(\frac{k \log^2 n \log(1/\delta) + k\sqrt{d \log(1/\delta)}}{\varepsilon}\right)$.

4: **Create** array $arr[1 : n]$.

5: **for** $i = 1$ **to** $n$ **do**

6:     $\tilde{x}_i := x_i + \frac{\rho \cdot \sqrt{2 \log(1.25/\delta)}}{\varepsilon} \cdot \mathcal{N}(0, I)$.

7:     **if** $d(\tilde{x}_i, F) \leq \rho \cdot S$ **then**

8:         $arr[i] = \arg\min_j d(\tilde{x}_i, f_j)$.

9:     **else**

10:         $\tilde{X}_0 = \tilde{X}_0 \cup \{\tilde{x}_i\}$

11: **for** $j = 1$ **to** $\alpha \cdot k$ **do**

12:     $\hat{X}_j = \{x_i : arr[i] = j\}$, and $\hat{n}_j = |\hat{X}_j|$.

13:     **if** $\hat{n}_j < T$ **then**

14:         $\tilde{X}_j$ is $\{\tilde{x}_i : arr[i] = j\}$.

15:     **else**

16:         Compute $\tilde{X}_j$ by applying a DP $k$-means (resp., $k$-median) semi-coreset algorithm (such as from Lemma A.10) to $\hat{X}_j$ with respect to $B(f_j, \rho \cdot S/\gamma)$, for some fixed $\gamma \leq \frac{1}{2}$.

17: $\tilde{X} = \bigcup_{\ell=0}^{m} \tilde{X}_\ell$

18: **Return** non-private $k$-means (resp., $k$-median) approximate solution with respect to $\tilde{X}$.

---

and s-sets is another non-geographic dataset and can be found in the clustering benchmark dataset[2]. For each dataset, we preprocess it to make it fit into $[-1, 1]^d$. We refer readers to Appendix E for more details of the preprocessing steps.

**Setup.** We compare our algorithm described in Algorithm 1 in Section 5 with other three algorithms. We report the $k$-means cost of all algorithms. In all plots, the label of our algorithm is "dist-DP $k$-means". The three compared baseline algorithms are as follows.

1. *Non-private baseline* ($k$-means++): We compare our algorithm with the non-private $k$-means solver using $k$-means++ seeding implemeted by Python scikit-learn package [66]. The output $k$-means cost of this baseline can be regarded as the groudtruth cost.

2. *DP baseline* (DP $k$-means): This is a $k$-means clustering algorithm in the standard DP setting implemented in part of a standard open-source DP library [3].

3. *$\rho$-Dist-DP baseline* (dist-DP random points): Finally, we also compare with a natural $\rho$-dist-DP algorithm described as the following. We run non-private $k$-means solver on $\tilde{X}$ described in Section 4. Since $\tilde{X}$ is a $\rho$-dist-DP version of $X$, the output centers are $\rho$-dist-DP. Note that since the final solution of this baseline only depends on $\tilde{X}$, we assign the entire privacy budget $(\varepsilon, \delta)$ to computing $\tilde{X}$.

In all experiments, we fix privacy parameters $\varepsilon = 1, \delta = 10^{-6}$. These parameter setups are standard in many other DP papers as well. We evaluate our algorithms for different choices of the privacy parameter $\rho$. Note that the parameter $\rho$ should not be determined by our algorithm. We try different $\rho$ to show how the choice of $\rho$ affects the clustering quality. We refer readers to Section 7 for more discussions of the choice of $\rho$.

We use the DP coreset implementation provided by the DP baseline for the purpose of the computation of semi-coreset $\tilde{X}_j$ described in Section 5.

**Our Results.** We run all algorithms for $k = 4, 6, 8, 12, 16$. For each experiment, we repeat 10 times and report the mean and the standard error. In the experiments shown in Figure 1, we fix $\rho = 0.05$[4]. As shown, the $k$-means cost of our dist-DP $k$-means algorithm is always smaller than the cost of DP

---

[2]https://cs.joensuu.fi/sipu/datasets/.

[3]https://ai.googleblog.com/2021/10/practical-differentially-private.html.

[4]We show advantages of our clustering for an example $\rho$ which neither depends on our algorithm nor be optimized. An example of the privacy guarantee of $\rho = 0.05$: For geographic (latitude and longitude) datasets (e.g., brightkite, gowalla), an attacker is hard to distinguish whether a user was in New York or in Toronto.

$k$-means baseline and is only slightly worse than the non-DP baseline which is as expected. The dist-DP baseline introduces a large $k$-means cost which implies that our partitioning strategies described in Section 4 and Section 5 are indeed necessary and can improve the clustering quality significantly in practice. Finally, we fix $k = 8$ and investigate how the changes of $\rho$ affect the $k$-means cost of our dist-DP $k$-means algorithm. We run our algorithm on all datasets for $\rho = 1, 0.08, 0.008, 0.0001$. As shown in Figure 2, the $k$-means cost of our algorithm decreases as $\rho$ decreases, which is as expected. For running time, though we did not optimize our implementation, each algorithm runs within at most a few minutes in a single thread mode.

In summary, for a reasonable range of $\rho$, we significantly outperform previous DP $k$-means algorithms, whereas more naive distance-based DP algorithms perform far worse. In addition, we have comparable approximation guarantees even to the non-private $k$-means algorithm.

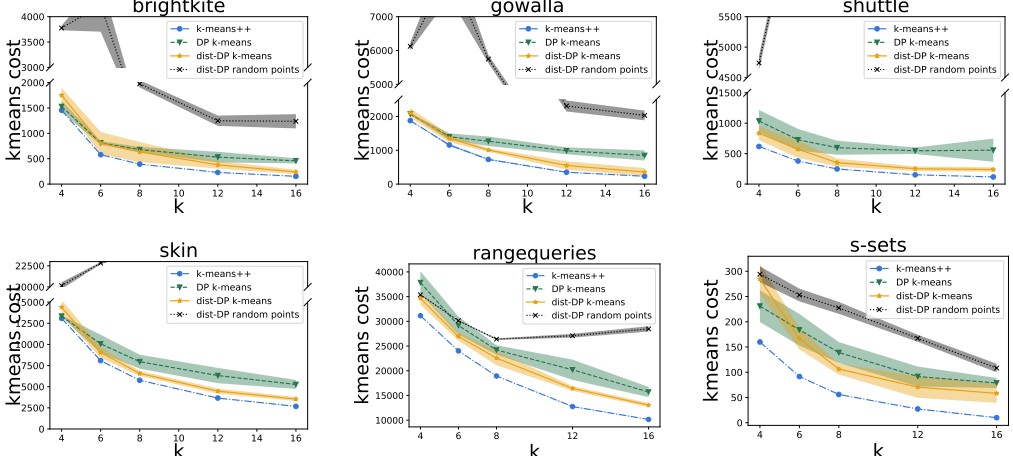

Figure 1: $k$-Means cost of non-private baseline (blue), DP baseline (green), our dist-DP $k$-means (yellow), and dist-DP baseline (gray) for different $k$ with $\rho = 0.05$. Shades indicate $3\times$ standard error over 10 runs.

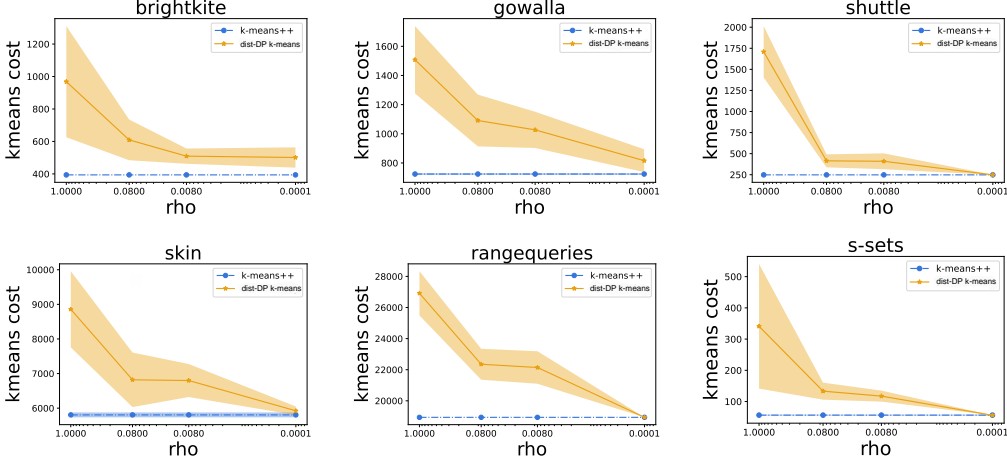

Figure 2: $k$-Means cost of dist-DP $k$-means algorithm for various $\rho$ with $k = 8$. Shades indicate $3\times$ standard error over 10 runs. The result supports the interpolating nature of the parameter $\rho$. In particular, when $\rho$ decreases, the $k$-means cost also decreases. When $\rho = 0$, we exactly recover the result as non-private $k$-means++.

# 7 Limitations and Open Problems

In this work, we propose efficient $(\varepsilon, \delta, \rho)$-dist-DP algorithms for $k$-means and $k$-median problems for any given privacy parameters $\varepsilon, \delta, \rho$. However, the choices of $\varepsilon, \delta$ and $\rho$ remain open. Notice that these privacy parameters should not be determined by our algorithm, but rather by legal teams, policy makers, or other experts for different specific scenarios. This is an expert determination that is outside of the scope of this paper but has been studied by practitioners extensively.

In proving Theorem 1.1, we obtains an additive error proportional to $k^2 \cdot \rho^2$ (ignoring polynomial factors in $d$ and logarithmic factors in the other parameters - see Theorem C.1), whereas the work of [61] has dependence $k \cdot \Lambda^2$. This is because to improve the dependence on $\Lambda$ to a dependence on $\rho$, we end up partitioning the data into roughly $k$ regions and must apply a separate private $k$-means algorithm on each region, which increases the additive dependence on $k$. Hence, a natural open question is whether one can improve the additive error's dependence on $k$.

**Acknowledgements:** SN is supported by a NSF Graduate Fellowship (Grant No. 1745302) and a Google PhD Fellowship.

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

# A    Additional Preliminaries

In this section, we state some additional definitions and preliminary results that are of use.

We note one notation that we may abuse: for positive reals $A, B$, we say that $C = A \pm B$ if $C \in [A - B, A + B]$. Likewise, we may say $C = (1 \pm \gamma)B$ if $C \in [(1 - \gamma)B, (1 + \gamma)B]$.

## A.1    Differential Privacy

In Section 2, we described the *Laplace mechanism* for approximating functions $f : X \to \mathbb{R}$.

Similar to the Laplace mechanism, there also exists the Gaussian mechanism, which is useful for high-dimensional functions $f : X \to \mathbb{R}^d$. We will state a simpler version that is sufficient for our purposes. Namely, a dataset consists of a single data point $x$, it is known that outputting $\tilde{x} = x + \rho \cdot \frac{\sqrt{2 \log(1.25/\delta)}}{\varepsilon} \cdot \mathcal{N}(0, I)$ satisfies $(\varepsilon, \delta, \rho)$-dist-DP, where $\mathcal{N}(0, I)$ represents a standard $d$-dimensional Gaussian. This implicitly follows from [32, Theorem 3.22]. As a result, we have the following basic proposition.

**Proposition A.1.** *Let $X = \{x_1, \dots, x_n\}$ be a dataset of size $n$. Then, the dataset $\{\tilde{x}_1, \dots, \tilde{x}_n\}$, where each $\tilde{x}_i$ is i.i.d. drawn as $x_i + \rho \cdot \frac{\sqrt{2 \log(1.25/\delta)}}{\varepsilon} \cdot \mathcal{N}(0, I)$, is $(\varepsilon, \delta, \rho)$-dist-DP.*

Next, we note two classic theorems regarding the privacy of composing private mechanisms (see, for instance, [32] or [72]). Note that these theorems hold for *adaptive* composition, which allows to run algorithms $\mathcal{A}_1, \dots, \mathcal{A}_k$ in sequence, where each $\mathcal{A}_i$ is allowed to treat $\mathcal{A}_1, \dots, \mathcal{A}_{i-1}$ as a fixed public input to its algorithm.

**Theorem A.2** (Basic Adaptive Composition). *Let $\mathcal{A}_1, \dots, \mathcal{A}_k$ be adaptive mechanisms on a dataset $X$ such that each $\mathcal{A}_i$ is $(\varepsilon_i, \delta_i)$-differentially private as a function of $X$, assuming that the previous outputs $\mathcal{A}_1, \dots, \mathcal{A}_{i-1}$ are fixed. Then, the mechanism $\mathcal{A}$ which concatenates the outputs of $\mathcal{A}_1, \dots, \mathcal{A}_k$ is $(\sum \varepsilon_i, \sum \delta_i)$-differentially private.*

*Likewise, if each $\mathcal{A}_i$ were $(\varepsilon_i, \delta_i, \rho)$-dist-DP, the concatenated mechanism $\mathcal{A}$ is $(\sum \varepsilon_i, \sum \delta_i, \rho)$-dist-DP.*

**Theorem A.3** (Advanced Adaptive Composition). *Let $\mathcal{A}_1, \dots, \mathcal{A}_k$ be adaptive mechanisms on a dataset $X$ such that each $\mathcal{A}_i$ is $(\varepsilon, \delta)$-differentially private as a function of $X$, assuming that the previous outputs $\mathcal{A}_1, \dots, \mathcal{A}_{i-1}$ are fixed. Then, for any $\delta' > 0$, the mechanism $\mathcal{A}$ which concatenates the outputs of $\mathcal{A}_1, \dots, \mathcal{A}_k$ is $(\sqrt{2k \log \delta^{-1}} \cdot \varepsilon + k\varepsilon(e^\varepsilon - 1), k\delta + \delta')$-differentially private.*

*Likewise, if each $\mathcal{A}_i$ were $(\varepsilon, \delta, \rho)$-dist-DP, the concatenated mechanism $\mathcal{A}$ is $(\sqrt{2k \log \delta^{-1}} \cdot \varepsilon + k\varepsilon(e^\varepsilon - 1), k\delta + \delta', \rho)$-dist-DP.*

## A.2    Clustering

In $k$-means or $k$-median clustering, given a dataset $X \subset B(0, \Lambda)$ of size $n$, we recall that our goal is to efficiently find a set of points $C$ such that $\mathrm{cost}(X; C)$ is a good approximation to $\mathrm{OPT}(X)$. In general, we wish for purely multiplicative approximations, but due to the nature of private $k$-means (and $k$-median), we will additionally have a small additive approximation that is proportional to $\Lambda^p$. We now define approximate $k$-means/$k$-median solutions.

**Definition A.4.** Suppose we are given data $X = \{x_1, \dots, x_n\} \in \mathbb{R}^d$, and implicit parameters $\rho$ and $k$. Then, for any $\beta \geq 1$, we define a set $C$ of size $k$ to be a $(\beta, V)$-*approximate solution for* $X$ if $\mathrm{cost}(X; C) \leq \beta \cdot \mathrm{OPT}(X) + V \cdot \rho^p$.

We also define bicriteria solutions for $k$-means and $k$-median: here, we are allowed to use a larger dataset $C$ that may have more than $k$ points, but still compare to the optimal $k$-clustering.

**Definition A.5.** Suppose we are given data $X = \{x_1, \dots, x_n\} \in \mathbb{R}^d$, and implicit parameters $\rho$ and $k$. Then, for any $\alpha, \beta \geq 1$, a set $C$ is an $(\alpha, \beta, V)$-*bicriteria approximate solution for* $X$ if $|C| \leq \beta \cdot k$ and $\mathrm{cost}(X; C) \leq \beta \cdot \mathrm{OPT}(X) + V \cdot \Lambda^p$.

Finally, we define *coresets* and *semi-coresets* for $k$-means (or $k$-median) clustering. A coreset of a dataset $X$, roughly speaking, is a (usually smaller) dataset $Y$ such that one can estimate a $k$-means

(or $k$-median) solution of $X$ by computing the solution on $Y$. More precisely, we have the following definition.

**Definition A.6.** Given a dataset $X = \{x_1, \ldots, x_n\}$ and some $\gamma, W \geq 0$, a $(\gamma, W, \rho)$-*coreset* for $k$-means (resp., $k$-median) is a dataset $\mathcal{Y}$ such that for *any* subset $C \subset \mathbb{R}^d$ of size $k$,

$$\frac{1}{1+\gamma} \cdot \text{cost}(Y, C) - W \cdot \rho^p \leq \text{cost}(Y; C) \leq (1+\gamma) \cdot \text{cost}(X, C) + W \cdot \rho^2,$$

where $p = 2$ (resp., $p = 1$). Likewise, a $(\kappa, W, \rho)$-*semi-coreset* (for $\kappa, W \geq 0$) for $k$-means (resp., $k$-median) is a dataset $\mathcal{Y}$ such that for *any* subset $C \subset \mathbb{R}^d$ of size $k$,

$$\frac{1}{1+\kappa} \cdot \text{cost}(Y, C) - \kappa \cdot \text{OPT}_k(X) - W \cdot \rho^p \leq \text{cost}(Y; C) \leq (1+\kappa) \cdot \text{cost}(X, C) + \kappa \cdot \text{OPT}_k(X) + W \cdot \rho^2.$$

### A.3 Randomly Shifted Grids

In our algorithms, we will make use of the Quadtree data structure which is composed of randomly shifted grids, which we now describe. This data structure has proven useful in various geometric settings beyond clustering, such as approximate near neighbor and computing other geometric quantities such as Earth-Mover distance and Minimum Spanning Tree cost.

**Definition A.7.** A *randomly shifted Quadtree* is constructed as follows. We start with a top level of some size $\Lambda$ and let level $0$ be a single grid cell, which is the $d$-dimensional hypercube $[-\Lambda, \Lambda]^d$. Next, we choose a uniformly random point $\nu = (\nu_1, \ldots, \nu_d) \in [-\Lambda, \Lambda]^d$, which will represent our shift vector. Now, for each level $\ell \geq 1$, we partition the region $[-\Lambda, \Lambda]^d$ into grid cells of size $\Lambda/2^\ell$, shifted by $\nu$. In other words, each cell is the form $[\nu_1 + a_1 \cdot \Lambda/2^\ell, \nu_1 + (a_1 + 1) \cdot \Lambda/2^\ell] \times \cdots \times [\nu_d + a_d \cdot \Lambda/2^\ell, \nu_d + (a_d + 1) \cdot \Lambda/2^\ell]$, where $a_1, \ldots, a_d \in \mathbb{Z}$. We say that $\Lambda/2^\ell$ is the *grid size* at level $\ell$. (We remark that we may truncate some grid cells so that they do not escape $[-\Lambda, \Lambda]^d$.) We continue this for a finite number of levels, until we reach some bottom level.

We will utilize the following fact about Quadtrees, or more specifically the randomly shifted grid at some fixed level $\ell$.

**Proposition A.8.** *(see Proof of Theorem B.1 in [25]) Given a randomly shifted grid of dimension* $20r \cdot d$, *a Euclidean ball of radius $r$ (in $\mathbb{R}^d$) is split into at most $2$ pieces in expectation.*

### A.4 Private $k$-means

Finally, we note the result of [61] on differentially private $k$-means (and $k$-median) clustering.

**Theorem A.9.** *[61] There exists a polynomial-time $(\varepsilon, \delta)$-DP algorithm that, given a set $X = \{x_1, \ldots, x_n\}$ in a fixed ball of radius $\Lambda$ in $\mathbb{R}^d$, outputs a set of $k$ centers $C = \{c_1, \ldots, c_k\}$ such that*

$$\text{cost}(X; C) \leq O(1) \cdot \text{OPT}_k(X) + U \cdot \Lambda^p,$$

*where $U = O\left(\frac{k \log^2 n \log(1/\delta) + k\sqrt{d \log(1/\delta)}}{\varepsilon}\right)$, and $p = 2$ for $k$-means and $p = 1$ for $k$-median.*

Using a slightly weaker result, [70, 25] was able to extend it to an algorithm for generating a private semi-coreset for $k$-means or $k$-median. Given Theorem A.9, the algorithm simply computes $C = \{c_1, \ldots, c_k\}$, and gives each $c_i$ a weight which is the number of points in $X$ closest to $c_i$, plus $\text{Lap}(1/\varepsilon)$ noise. By combining Theorem A.9 and the conversion of [70, 25] (e.g., see [25, Lemma C.1], which uses a slightly weaker bound), the following is immediate.

**Lemma A.10.** *For some $\kappa = O(1)$, there exists a polynomial-time $(\varepsilon, \delta)$-DP algorithm that, given a set $X = \{x_1, \ldots, x_n\}$ in a fixed ball of radius $R$ in $\mathbb{R}^d$, computes a $(\kappa, U, \rho)$-semi-coreset for $k$-means, where $U = O\left(\frac{k \log^2 n \log(1/\delta) + k\sqrt{d \log(1/\delta)}}{\varepsilon}\right)$.*

## B Crude Approximation

In this section, we devise a crude bicriteria approximation that will serve as a starting point in developing our more refined algorithm. To recall the setup of the bicriteria problem (see Definition

---

**Algorithm 2** Approximate dist-DP bicriteria algorithm

---

1: **Input:** Parameters $n, d, k, \varepsilon, \delta, \Lambda, \rho$, dataset $X = \{x_1, \ldots, x_n\} \subset \mathbb{R}^d$.
2: **Output:** Crude bicriteria approximation $F = \{f_1, \ldots, f_{\alpha \cdot k}\} \subset \mathbb{R}^d$: will be $(O(\varepsilon), O(\delta), \rho)$-dist-DP.
3: **Initialize** $A = O(\varepsilon^{-1}\sqrt{\log \delta^{-1}} \cdot d\sqrt{d + \log n})$, $B = n$, $REP = O(\log n)$.
4: **Initialize** $\varepsilon' = \Theta\left(\varepsilon/\sqrt{\log n \log(A \cdot B) \log(1/\delta)}\right)$ and $\delta' = \Theta\left(\delta/(\log n \log(A \cdot B))\right)$.
5: **for** $i = 1$ **to** $n$ **do**
6:     $\tilde{x}_i := x_i + \frac{\rho \cdot \sqrt{2\log(1.25/\delta)}}{\varepsilon} \cdot \mathcal{N}(0, I)$.
7: **for** $rep = 1$ **to** $REP$ **do**
8:     **Create** a randomly shifted Quadtree with largest level $\ell = 0$ with side length $\Lambda$ and smallest level with side length $\rho/B$.
9:     **for** $\ell = 0$ **to** $L_1 := \log_2(\Lambda/(A\rho))$ **do**
10:       **for** each cell $g$ at level $\ell$ containing some $\tilde{x}_i$ **do**
11:         $count(g) = \#\{\tilde{x}_i \text{ in cell } g\}$.
12:       **Let** $g_1, \ldots, g_{4k}$ be the $4k$ cells at level $\ell$ with maximum $count(g)$.
13:       **Add** the centers of $g_1, \ldots, g_{4k}$ to $F$.
14:     **for** $\ell = \log_2\left(\frac{\Lambda}{A\rho}\right) + 1$ **to** $L_2 := \log_2\left(\frac{B\Lambda}{\rho}\right)$ **do**
15:       **for** each cell $g$ at level $\ell$ that contains some $x_i \in X$ **do**
16:         $count(g) = \#\{x_i \text{ in cell } g\} + \text{TLap}(1/\varepsilon', 1/\delta')$.
17:       **Let** $g_1, \ldots, g_{4k}$ be the $4k$ cells at level $\ell$ with maximum $count(g)$.
18:       **Add** each center $g_i$ to $F$, if $count(g_i) \geq \frac{K}{\varepsilon'}\log\frac{1}{\delta'}$ for some constant $K$.
19: **Return** $F$.

---

A.5), we are given a dataset $X = \{x_1, \ldots, x_n\}$, contained in a given ball of radius $\Lambda$ in $\mathbb{R}^d$. We wish to compute an $(\alpha, \beta, V)$-approximation that satisfies $(\varepsilon, \delta, \rho)$-dist-DP. By this, we must output a set of $\alpha \cdot k$ centers $F = \{f_1, \ldots, f_{\alpha \cdot k}\}$ such that $\text{cost}(X; F) \leq \beta \cdot \text{OPT}_k(X) + V \cdot \rho^p$ for some parameters $\alpha, \beta, V$, where $p = 1$ for $k$-median and $p = 2$ for $k$-means. In addition, $F$ should be $(\varepsilon, \delta, \rho)$-dist-DP with respect to $X$.

Our desire for $\alpha, \beta, V$ is that they are polynomial in $d, k, \log n, \varepsilon^{-1}, \log \delta^{-1}$, and $\log \frac{\Lambda}{\rho}$. We do not wish for any polynomial dependencies on $n$, either in the approximation ratio or in the additive error.

We recall the algorithm description from Section 4. We also include the pseudocode here, as Algorithm 2.

We now focus on analyzing the privacy and accuracy of the algorithm. Formally, in this section we prove the following.

**Theorem B.1.** *For any* $0 < \varepsilon, \delta < \frac{1}{2}$ *and* $\rho \leq \frac{\Lambda}{2}$, *there exists an* $(\varepsilon, \delta, \rho)$-*dist-DP* $(\alpha, \beta, V)$-*bicriteria approximation for k-median, with* $\alpha = O\left(\log n \cdot (\log n + \log \frac{\Lambda}{\rho})\right)$, $\beta = O(d^{3/2})$, *and* $V = kd^2/\varepsilon^2 \cdot \text{poly}\log(n, d, \varepsilon^{-1}, \delta^{-1})$.

*Likewise, there exists an* $(\varepsilon, \delta, \rho)$-*dist-DP* $(\alpha, \beta, V)$-*bicriteria approximation for k-means, with* $\alpha = O\left(\log n \cdot (\log n + \log \frac{\Lambda}{\rho})\right)$, $\beta = O(d^3)$, *and* $V = kd^4/\varepsilon^3 \cdot \text{poly}\log(n, d, \varepsilon^{-1}, \delta^{-1})$.

**Analysis of Privacy:** The $\tilde{x}_i$ points will be $(\varepsilon, \delta, \rho)$-dist-DP, by the Gaussian Mechanism (Proposition A.1). The levels above $\rho \cdot A$ are strictly determined by $\tilde{x}_i$. For each level of grid length between $\rho \cdot A$ and $\rho/B$, changing one data point changes at most 2 grid cells each by 1, which implies $(2\varepsilon', 2\delta')$-DP. In addition, this happens over $O(\log(A \cdot B))$ levels and $O(\log n)$ repetitions for each. By applying the advanced composition theorem (Theorem A.3, we have that as long as $\varepsilon, \delta < 1$, for our choices of $\varepsilon', \delta'$, the composition is $(O(\varepsilon), O(\delta))$-DP.

In total, the algorithm is $(O(\varepsilon), O(\delta), \rho)$-dist-DP.

**Analysis of Accuracy:** Let $X = \{x_1, \ldots, x_n\}$ be our original set of points, and let $C = \{c_1, \ldots, c_k\}$ be the optimal set of $k$ centers. For any radius $r$, let $n_r$ be the number of points $x \in X$ such that $d(x, C) \geq r$. Then, it is well known that the $k$-means cost and $k$-median cost, up to an $O(1)$-

multiplicative factor, equal

$$\sum_{t\in\mathbb{Z}} 2^{2t} \cdot n_{2^t} \quad \text{and} \quad \sum_{t\in\mathbb{Z}} 2^t \cdot n_{2^t},$$

respectively. For the set of centers $F$ generated, we similarly define $\hat{n}_r$ to be the number of points $x \in X$ such that $d(x, C) \geq r$.

It is well-known that the magnitude of a $d$-dimensional Gaussian $\mathcal{N}(0, I)$ is bounded by $O(\sqrt{d + \log 1/\beta})$ with failure probability $\beta$. Hence, we set $A = O(\varepsilon^{-1}\sqrt{\log \delta^{-1}} \cdot d\sqrt{d + \log n})$, so that with high probability, $\|\tilde{x}_i - x_i\|_2 \leq \rho \cdot A/(40d)$ for all $i$. Now, for any $r \geq \rho \cdot A/(40d)$, if there exist $k$ balls of radius $r$ that contain all but $n_r$ of the points in $X$, then there exist $k$ balls of radius $2r$ that contain all but $n_r$ of the points in $\tilde{X}$. In addition, given a randomly shifted grid of dimension $40r \cdot d$, a ball of radius $2r$, in expectation, is split into at most 2 pieces, by Proposition A.8. Therefore, by Markov's inequality, the $k$ balls of radius $2r$ are split into at most $4k$ cells with at least $50\%$ probability, which means that the top $4k$ cells at grid level $40rd$ contain all but at most $n_r$ points. Hence, because the center of the $40r \cdot d$-side length grid has radius $20rd^{3/2}$, this means $\hat{n}_{20rd^{3/2}} \leq n_r$ for all $r \geq \rho \cdot A/(40d)$ with at least $50\%$ probability: repeating this $O(\log n)$ times, this holds with at least $1 - n^{-5}$ probability, even across all levels.

Next, suppose that $r \leq \rho \cdot A/(40d)$. In this case, if we didn't add noise we would have $\hat{n}_{20rd^{3/2}} \leq n_r$ as in the previous case. This time, however, we add noise to the count of each cell rather than the number of points. In addition, we may not include a cell if its noisy count is at most $\frac{K}{\varepsilon'} \cdot \log \frac{1}{\delta'}$, but note that this means its true count is at most $\frac{2K}{\varepsilon'} \cdot \log \frac{1}{\delta'}$. Therefore, since the count of each cell is altered by $O(\frac{1}{\varepsilon'} \log \frac{1}{\delta'})$, we have that $\hat{n}_{20rd^{3/2}} \leq n_r + O(\frac{k}{\varepsilon'} \cdot \log \frac{1}{\delta'})$ for all $r \leq \rho \cdot \frac{A}{40d}$.

In summary, we have that $\hat{n}_{20rd^{3/2}} \leq n_r$ for $r \geq \rho \cdot O(\varepsilon^{-1}\sqrt{\log \delta^{-1}} \cdot \sqrt{d + \log n})$. In addition, for $r \leq \rho \cdot O(\varepsilon^{-1}\sqrt{\log \delta^{-1}} \cdot \sqrt{d + \log n})$, we have that $\hat{n}_{20rd^{3/2}} \leq n_r + O(k/\varepsilon' \cdot \log 1/\delta')$. If we set $B = n$, then below $r = \rho \cdot \sqrt{d}/n$ we have $\hat{n}_r \leq n$ by default and above $r = \rho \cdot \sqrt{d}/n$ the above bounds hold. This implies that

$$\sum_{t\in\mathbb{Z}} 2^t \hat{n}_{2^t} \leq O(d^{3/2}) \cdot \left(\sum_{t\in\mathbb{Z}} 2^t n_{2^t}\right) + O(\rho) \cdot d^{3/2} \cdot \sum_{t\in\mathbb{Z}:2^t \leq \varepsilon^{-1}\sqrt{\log \delta^{-1}}\cdot\sqrt{d+\log n}} 2^t \cdot O\left(\frac{k}{\varepsilon'} \cdot \log \frac{1}{\delta'}\right) + O\left(\rho \cdot \frac{\sqrt{d}}{n}\right) \cdot n$$

$$= O(d^{3/2}) \cdot \text{OPT}_k(X) + O\left(\frac{kd^2}{\varepsilon^2}\right) \cdot \text{poly}\log(n, d, \varepsilon^{-1}, \delta^{-1}) \cdot \rho.$$

Hence, we obtain a bicriteria with multiplicative approximation $\beta = O(d^{3/2})$ and additive error $k\sqrt{d} \cdot \text{poly}(\varepsilon^{-1}, \log \delta^{-1}, \log n) \cdot \rho$, for $k$-median. The same calculation for $k$-means will give us a multiplicative approximation $\beta = O(d^3)$ and additive error $kd^4/\varepsilon^3 \cdot \text{poly}\log(n, d, \varepsilon^{-1}, \delta^{-1}) \cdot \rho^2$.

Finally, the number of centers we output is simple to compute. We have $O(\log n)$ repetitions, and each repetition has $O\left(\log \frac{\Lambda}{\rho/n}\right)$ levels, each of which we select at most $4k$ cell centers from. Hence, we select $O\left(k \cdot \log n \cdot (\log n + \log \frac{\Lambda}{\rho})\right)$ points, meaning that $\alpha = O\left(\log n \cdot (\log n + \log \frac{\Lambda}{\rho})\right)$.

## C   From Crude to Accurate

In this section, we devise an improved approximation that only uses $k$ centers and achieves a constant approximation ratio. We will subsequently prove Theorem 1.1.

Our approach utilizes both the crude approximation from Section 4/Section B and previously known constant-approximation differentially private (but not dist-DP) algorithms from the literature. We show how to combine these to create a dist-DP semi-coreset. This idea is partially inspired by the work of [22] for (non-private) coreset constructions and more recently [25] for fast private (semi-)coreset constructions, More accurately, given a set of $n$ points $X = \{x_1, \ldots, x_n\} \in \mathbb{R}^d$, we will compute a (weighted) set of points $Y$ that is $(\varepsilon, \delta, \rho)$-dist-DP with respect to $X$, such that for any set of $k$ centers $C = \{c_1, \ldots, c_k\}$, $\text{cost}(Y; C) = \Theta(\text{cost}(X; C)) \pm O(\text{OPT}_k(X)) \pm W \cdot \rho^p$, where $W$ will be polynomial in $d, k, \varepsilon^{-1}, \log \delta^{-1}, \log n$, and $\log \frac{\Lambda}{\rho}$.

If we can achieve this, then we just have to compute an $O(1)$-approximate $k$-means (or $k$-median) solution to $Y$, which does not have to be private since $Y$ already is. Indeed, if we do so, then the centers $C$ that we find for $Y$ satisfy

$$\text{cost}(Y;C) \leq O(1) \cdot \text{cost}(Y;C^*) \leq O(1) \cdot \text{cost}(X;C^*) + O(\text{OPT}_k(X)) + O(W) \cdot \rho^p$$

for any set of $k$ centers $C^*$. Hence, this implies that

$$\text{cost}(Y;C) \leq O(1) \cdot \text{OPT}_k(X) + O(W) \cdot \rho^p.$$

Finally, we also have that $\text{cost}(Y;C) \geq \Omega(1) \cdot \text{cost}(X;C) - O(\text{OPT}_k(X)) - W \cdot \rho^p$, which means $\text{cost}(X;C) \leq O(1) \cdot \text{cost}(Y;C) + O(\text{OPT}_k(X)) + O(W) \cdot \rho^p$, so as desired, we have

$$\text{cost}(X;C) \leq O(1) \cdot \text{OPT}_k(X) + O(W) \cdot \rho^p.$$

Hence, it suffices to prove the following theorem.

**Theorem C.1.** *For any $0 < \varepsilon, \delta < \frac{1}{2}$ and $\rho \leq \frac{\Lambda}{2}$, there exists an $(\varepsilon, \delta, \rho)$-dist-DP $(O(1), W, \rho)$-semi-coreset for $k$-means (resp., $k$-median), with $W = O\left(\frac{k^2 d^{4.5}}{\varepsilon^3}\right) \cdot \text{poly} \log\left(n, d, \frac{1}{\varepsilon}, \frac{1}{\delta}, \frac{\Lambda}{\rho}\right)$ for $k$-means and $W = O\left(\frac{k^2 d^{2.5}}{\varepsilon^2}\right) \cdot \text{poly} \log\left(n, d, \frac{1}{\varepsilon}, \frac{1}{\delta}, \frac{\Lambda}{\rho}\right)$ for $k$-median.*

We assume we have a private $(\alpha, \beta, V)$-bicriteria approximation $F$. Recall this means we have an $(\varepsilon, \delta, \rho)$-dist-DP set of (at most) $\alpha \cdot k$ centers $F$ such that $\text{cost}(X;F) \leq \beta \cdot \text{OPT}_k(X) + V \cdot \rho^p$.

We recall the algorithm description and pseudocode (Algorithm 1) from Section 5. Hence, for the remainder of this section we focus on proving Theorem C.1. In addition, after proving Theorem C.1, we briefly discuss the runtime and how to make the runtime close to linear, and parallelizable.

**Analysis of Privacy:** We will think of the algorithm as having 3 adaptive components. First, we must create $F$, which is $(\varepsilon, \delta, \rho)$-dist-DP, by Theorem B.1. In addition, we create $\{\tilde{x}_i\}$, which as a set is $(\varepsilon, \delta, \rho)$-dist-DP, by Proposition A.1. Note that $\tilde{X}_0$, the sets $I_j$, and the sizes $\hat{n}_j$ are also only depend on $F$ and $\{\tilde{x}_i\}$. So, for each $j$ with $\hat{n}_j < T$, the algorithm's creation of $\tilde{X}_j$ is only depends on $F$ and $\{\tilde{x}_i\}$. Finally, we must compute $\tilde{X}_j$ for each $j$ such $\hat{n}_j \geq T$. However, each coreset is $(\varepsilon, \delta)$-DP which also implies $(\varepsilon, \delta, \rho)$-dist-DP, and we are computing the coresets on disjoint subsets of indices, which are fixed. So overall, computing all of the $\tilde{X}_j$ is $(\varepsilon, \delta, \rho)$-dist-DP if we fix $F$ and each $\tilde{x}_i$.

By basic adaptive composition (Theorem A.2), the overall procedure is $(3\varepsilon, 3\delta, \rho)$-dist-DP.

**Analysis of Accuracy:** We focus on accuracy for $k$-means; the proof for $k$-median is extremely similar.

First, note that $d(x_i, \tilde{x}_i) \leq O\left(\frac{\rho\sqrt{\log(1/\delta)}}{\varepsilon} \cdot \sqrt{d + \log n}\right)$ for all $i$. Also, note that for general positive reals $A, B$, $(A \pm B)^2 = A^2 \pm 2AB + B^2$, and $2AB \leq \gamma A^2 + \frac{1}{\gamma} B^2$ for any positive $\gamma$. This means that for any $0 < \gamma < 1$, $(A+B)^2 = (1 \pm \gamma)A^2 \pm O\left(\frac{1}{\gamma}\right) B^2$, and $(A-B)^2 = (1 \pm \gamma)A^2 \pm O\left(\frac{1}{\gamma}\right) B^2$. Hence,

$$d(\tilde{x}_i, C)^2 = \left(d(x_i, C) \pm O\left(\frac{\sqrt{\log(1/\delta)}}{\varepsilon} \cdot \sqrt{d + \log n}\right) \cdot \rho\right)^2$$

$$= (1 \pm \gamma) \cdot d(x_i, C)^2 \pm O\left((d + \log n) \cdot \frac{\log(1/\delta)}{\varepsilon^2} \cdot \frac{1}{\gamma}\right) \cdot \rho^2. \qquad (1)$$

Therefore, for any set $C$ of size at most $k$,

$$\sum_{i \in I_0} d(\tilde{x}_i, C)^2 = (1 \pm \gamma) \cdot \sum_{i \in I_0} d(x_i, C)^2 \pm O\left((d + \log n) \cdot \frac{\log(1/\delta)}{\varepsilon^2} \cdot \frac{1}{\gamma}\right) \cdot \rho^2 \cdot |I_0|,$$

where we recall that $i \in I_0$ if and only if $d(\tilde{x}_i, F) > S \cdot \rho$. For $S \geq \Omega\left(\frac{\sqrt{\log(1/\delta)}}{\varepsilon} \cdot \sqrt{d + \log n}\right)$, this implies that $d(x_i, F) \geq \frac{S}{2} \cdot \rho$. However, if $d(\tilde{x}_i, F) > S \cdot \rho$, then $d(x_i, F) \geq \frac{S}{2} \cdot \rho$, and the

number of such $i$ with $d(x_i, F) \geq \frac{S}{2} \cdot \rho$ is at most $\frac{\text{cost}(X;F)}{(S/2)^2\rho^2} \leq \frac{4\beta \cdot \text{OPT}_k(X) + 4V \cdot \rho^2}{S^2 \cdot \rho^2}$. Hence, apart from the $1 \pm \gamma$ multiplicative error, we also incur an additional additive error of

$$O\left((d + \log n) \cdot \frac{\log(1/\delta)}{\varepsilon^2} \cdot \frac{1}{\gamma}\right) \cdot \frac{\beta}{S^2} \cdot \text{OPT}_k(X) + O\left((d + \log n) \cdot \frac{\log(1/\delta)}{\varepsilon^2} \cdot \frac{1}{\gamma}\right) \cdot \frac{V}{S^2} \cdot \rho^2.$$

So, by setting $S = O\left(\frac{1}{\varepsilon \cdot \gamma} \cdot \sqrt{(d + \log n) \cdot \log(1/\delta) \cdot \beta}\right)$, we obtain an additional $1 \pm O(\gamma)$ multiplicative error and an additive error of $O\left(\frac{\gamma}{\beta} \cdot V \cdot \rho^2\right)$. This deals with the error from points sent to $\tilde{X}_0$. To summarize, we have

$$\sum_{i \in I_0} d(\tilde{x}_i, C)^2 = (1 \pm O(\gamma)) \cdot \sum_{i \in I_0} d(x_i, C)^2 \pm O\left(\frac{\gamma}{\beta} \cdot V \cdot \rho^2\right) \tag{2}$$

for all sets $C$ of size at most $k$.

Next, we deal with points in $\hat{X}_j$ with $\hat{n}_j < T$. In this case, we still have that (1) holds, which means for any such $j$ and any subset $C$ of $k$ points,

$$\sum_{i \in I_j} d(\tilde{x}_i, C)^2 = (1 \pm \gamma) \cdot \sum_{i \in I_j} d(x_i, C)^2 \pm O\left((d + \log n) \cdot \frac{\log(1/\delta)}{\varepsilon^2} \cdot \frac{1}{\gamma} \cdot T\right) \cdot \rho^2, \tag{3}$$

since $|I_j| = \hat{n}_j < T$.

Finally, we deal with the rest of the points, for which we use a regular differentially private semi-coreset algorithm on each $\hat{X}_j$. Recall that we choose $S$ so that $\|\tilde{x}_i - x_i\|_2 \leq S \cdot \rho$ for all $i$, which means every point $x_i$ for $i \in I_j$ is in $B(f_j, 2S\rho)$, i.e., the ball of radius $2S\rho$ of $f_j$. We apply the private semi-coreset algorithm from Lemma A.10 with respect to the larger ball $B(f_j, \rho \cdot S/\gamma)$. This means that for any subset $C$ of size at most $k$ in $\mathbb{R}^d$,

$$\text{cost}(\tilde{X}_j; C) = \Theta(1) \cdot \text{cost}(\hat{X}_j; C) \pm O(1) \cdot \text{OPT}_k(\hat{X}_j) \pm U \cdot \left(\frac{2S\rho}{\gamma}\right)^2, \tag{4}$$

where $U = O\left(\frac{k \log^2 n \log(1/\delta) + k\sqrt{d \log(1/\delta)}}{\varepsilon}\right)$. We emphasize that Lemma A.10 holds even with respect to a center set $C$ that is *not contained* in the ball $B(f_j, \rho \cdot S/\gamma)$.

We now combine Equations (2), (3), and (4), setting $\gamma$ to be a fixed small constant. Since $\tilde{X}$ is the aggregation of all $\tilde{X}_j$'s for $j = 0, 1, \ldots, \alpha \cdot k$, and recalling that $\hat{X}_j = \{x_i : i \in I_j\}$, we have that

$$\text{cost}(\tilde{X}; C)$$
$$= \sum_{j=0}^{\alpha \cdot k} \text{cost}(\tilde{X}_j; C)$$
$$= \Theta(1) \cdot \sum_{j=0}^{\alpha \cdot k} \text{cost}(\hat{X}_j; C) \pm O(1) \cdot \sum_{j=1}^{\alpha \cdot k} \text{OPT}_k(\hat{X}_j) \pm O\left(\frac{V}{\beta} + \alpha k \cdot (d + \log n) \cdot \frac{\log(1/\delta)}{\varepsilon^2} \cdot T + \alpha k \cdot U \cdot S^2\right) \cdot \rho^2$$
$$= \Theta(1) \cdot \text{cost}(X; C) \pm O(1) \cdot \text{OPT}_k(X) \pm O\left(\frac{k^2 d^{4.5}}{\varepsilon^3}\right) \cdot \text{poly} \log\left(n, d, \frac{1}{\varepsilon}, \frac{1}{\delta}, \frac{\Lambda}{\rho}\right) \cdot \rho^2.$$

The second line is true by combining the equations and setting $\gamma$ to be a small constant. The third line is true since $\hat{X}_j$ forms a partition of $X$, and by our parameter settings of $\alpha, \beta, S, T, U, V$.

This completes the proof of Theorem C.1, in the $k$-means case. The $k$-median case follows the same analysis (though we will set $S = O\left(\frac{1}{\varepsilon \cdot \gamma} \cdot \sqrt{(d + \log n) \cdot \log(1/\delta) \cdot \beta}\right)$ in this case), and will result in the additive term of $O\left(\frac{V}{\beta} + \alpha k \cdot \sqrt{(d + \log n) \cdot \frac{\log(1/\delta)}{\varepsilon^2}} \cdot T + \alpha k \cdot U \cdot S\right) \cdot \rho = O\left(\frac{k^2 d^{2.5}}{\varepsilon^2}\right) \cdot \text{poly} \log\left(n, d, \frac{1}{\varepsilon}, \frac{1}{\delta}, \frac{\Lambda}{\rho}\right) \cdot \rho$. By combining this with our discussion at the beginning of this section, we have also proven Theorem 1.1.

**Runtime:** Finally, we note that this algorithm can be implemented efficiently. Indeed, in Algorithm 2, creating the points $\tilde{x}_i$, creating each quadtree data structure, computing the counts (only for the $\tilde{x}_i$ and $x_i$ points), and adding laplace noise all takes $\tilde{O}(nd)$ time, where we also hide logarithmic factors in $\frac{\Lambda}{\rho}$. Finally, picking the $k$ heaviest cells in each grid also takes $\tilde{O}(n)$ time.

In Algorithm 1, there are two potential bottlenecks. The first is mapping each point $\tilde{x}_i$ to its closest center $f_j$, which takes $O(nd \cdot |F|) = \tilde{O}(ndk)$ time, hiding logarithmic factors in $\frac{\Lambda}{\rho}$. We additionally have to compute a private semi-coreset for the points in each $\alpha \cdot k$ sets of points $\hat{X}_j$. However, using the private algorithm of [25, Theorem C.2], computing an $O(1)$-approximate private semi-coreset can be done in time $\tilde{O}(\hat{n}_j d) + \text{poly}(k) \cdot d$. Note that $\hat{n}_j = |\hat{X}_j|$ and $\sum \hat{n}_j = n$. Hence, because $\alpha = O\left(\log \frac{\Lambda}{\rho}\right)$, the overall algorithm, apart from the assignment of each $x_i$ to $\hat{X}_j$, takes $\tilde{O}(nd) + \text{poly}(k) \cdot d$ time, hiding logarithmic factors in $\frac{\Lambda}{\rho}$.

To improve the $nkd$ to $nd$, we may use a $K = O(\log n)$-approximate nearest neighbor data structure to map each $\tilde{x}_i$ to its $K$-approximate nearest neighbor $f_j \in F$. By using the locality-sensitive hashing algorithm of [4], we can compute every $K$-approximate nearest neighbor of each $\tilde{x}_i$ in $\tilde{O}(nd)$ time instead. We remark that the privacy analysis will be unchanged, and the accuracy analysis will be similar, up to getting a slightly worse additive approximation. Namely, if the points $\hat{X}_j$ were previously within $O(S)$ of the center $f_j$, they may now have distance $O(S \cdot K)$. Hence, the semi-coreset computation will have to be done with respect to a ball of radius $O(\rho \cdot S \cdot K/\gamma)$, but for $K = O(\log n)$ and $\gamma$ a constant, this doesn't affect the additive error by more than an $O(\log^2 n)$ factor.

Hence, we can compute a private semi-coreset in $\tilde{O}(nd) + \text{poly}(k) \cdot d$ time. Finally, we need to compute an offline (non-private) $k$-means (or $k$-median) approximation. As this is not related to private clustering, we simply sketch how this can be done.

First, in linear ($\tilde{O}(nd)$) time, the method of [22] computes an $O(1)$-approximate coreset $C$ of size $\text{poly}(k, \log n) \cdot d$. We can then project the data onto $O(\log k)$ dimensions, with a linear map $\Pi$. In low dimensions, we can compute a smaller coreset $C'$ of size $\text{poly}(k, \log n)$ of $\Pi C$ in linear time, and then solve $k$-means on $C'$ in time $\text{poly}(k, \log n)$. This also implies an $O(1)$-approximation for $\Pi C$. Next, we can map every point in $\Pi C$ to its closest center in $d' = O(\log k)$ dimensions, to form an explicit clustering. This takes time $O(|C| \cdot k \cdot d') = \text{poly}(k, \log n) \cdot d$ time. By [57], every $k$ clustering has its $k$-means objective preserved by a $\Theta(1)$-approximate when projected by $\Pi$, which means the same clustering should still be an $O(1)$-approximation in the original space. We can compute the mean of each cluster in linear time, so in the original $d$-dimensional space, we can find an $O(1)$-approximate $k$-means clustering in time $\tilde{O}(nd) + \text{poly}(k) \cdot d$, as desired. Finally, in the $k$-median case, [57] is still applicable, and we can compute an approximate 1-median of each clustering in near-linear time as well [23].

**Parallel computation.** In the following, we briefly discuss how to implement our algorithm in the massively parallel computation (MPC) model [51, 11] when each machine has $(kd \log(n) \cdot 1/\varepsilon \cdot \log 1/\delta \cdot \log(\Lambda/\rho))^C$ memory for a sufficiently large constant $C > 0$. Before we state our implmentation, let us briefly describe the MPC model. In the MPC model, there are $M$ machines where each machine has $H$ local memory where $H$ is sublinear in the input size and $H = M^\gamma$ for an arbitrary constant $\gamma > 0$. At the beginning of the computation, the input is arbitrarily distributed in the machines. The computation proceeds in rounds. In each round, each machine performs some local computation. Then at the end of the round, each machine sends/recieves messages to/from other machines. However, the messages sent/recived by a machine in a round cannot exceed its local memory $H$. At the end of the algorithm, the output should stored in machines distributedly. The goal is to design an algorithm with small number of rounds. In the following, we show how to implement our algorithm in the MPC model using $O(1)$ rounds.

Consider Algorithm 2. Computation of $\{\tilde{x}_1, \tilde{x}_2, \cdots, \tilde{x}_n\}$ only requires local computations. Then, we can run $REP$ repetitions and each level $\ell \in [0, L_2]$ of the loop in Algorithm 2 in parallel. For each instance, it only requires the counting and taking maximum which can be easily done in the MPC model in $O(1)$ rounds [45]. Since a single machine has large enough local memory, we are able to send the entire $F$ to a single machine. The total space required here is $O(REP \cdot L_2 \cdot n \cdot d)$. Next, let

us consider the implementation of Algorithm 1. Since each machine has enough local memory, we are able to make each machine holds a copy of $F$. This broadcasting process can be done in $O(1)$ rounds (see e.g., [5]). Once each machine holds $F$, only local computation is required to determine a point $\tilde{x}_i$ whether it should be in $\tilde{X}_0$. For points that are not belong to $\tilde{X}_0$, we are able to determine whether it belongs to $\hat{X}_j$ by only local computations. For each $\hat{X}_j$, we run the MPC DP coreset algorithm of [25] to get an semi-coreset. This step also takes $O(1)$ rounds. Finally, we can run any non-private MPC $k$-means algorithm (e.g., [37]) on $\tilde{X}_0 \cup \tilde{X}_1 \cdots \cup \tilde{X}_{|F|}$ which takes $O(1)$ rounds.

## D  A simple lower bound for dist-DP clustering

In this section, we prove the following simple proposition, showing a additive dependence on $k \cdot \rho$ (resp., $k \cdot \rho^2$) is necessary for $k$-median (resp, $k$-means), as long as the dimension is $d = \Omega(\log k)$.

**Proposition D.1.** *Let $X_0 = \{x_1, \ldots, x_{2k}\}$ be points in the ball of radius $\rho$ around the origin, separated by at least $\frac{\rho}{10}$ (for $d = \Omega(\log k)$, this is doable). Suppose $X \subset X_0$ is a random subset of size $k$, and there exists an $(\varepsilon, \delta, \rho)$-dist-DP algorithm that outputs $k$ centers $C$ in terms of $X$, where $\varepsilon, \delta \leq 0.1$. Then, the expected cost $\mathbb{E}[\text{cost}(X; C)]$ is $\Omega(k \cdot \rho^p)$, where $p = 1$ for $k$-median and $p = 2$ for $k$-means. Yet, the optimum cost $\text{OPT}_k(X)$ is $0$.*

*Hence, any $(\varepsilon, \delta, \rho)$-dist-DP algorithm with finite multiplicative ratio must incur additive error $k \cdot \rho^p$.*

*Proof.* First, note that $\text{OPT}_k(X) = 0$ since $|X| = k$, so for $C^* = X$, $\text{cost}(X, C^*) = 0$. We now show that $\mathbb{E}[\text{cost}(X; C)] = \Omega(k \cdot \rho^p)$.

For each $i \leq 2k$, let $B_i$ be the ball of radius $\frac{\rho}{100}$ around $x_i$. Let $p_i$ be the probability over $X$ and the randomness of the private algorithm that some point in $C$ is in $B_i$. Let $p_i^+$ be the probability of the same event conditioned on $x_i \in X$, and $p_i^-$ be the same probability conditioned on $x_i \notin X$.

First, note that $x_i \in X$ with probability $1/2$, since $|X| = \frac{1}{2} \cdot |X_0|$. So, $p_i = \frac{1}{2}(p_i^- + p_i^+)$. Next, there exists a simple coupling between the events of $x_i \in X$ and $x_i \notin X$, that changes at most 1 point. Namely, if $X$ contains $x_i$, add in a random point in $X_0 \backslash X$, and then remove $x_i$, to get a new set $X'$. If the distribution of $X$ is uniform conditioned on $x_i \in X$, it is simple to see that the distribution of $X'$ is uniform conditioned on $x_i \notin X$. Therefore, $\mathbb{P}(C(X) \in B_i) = p_i^+$ and $\mathbb{P}(C(X') \in B_i) = p_i^-$.

Because we only changed one element $x_i$ and moved it a distance at most $\rho$, this means that $\mathbb{P}(C(X) \in B_i) = e^{\pm\varepsilon} \cdot \mathbb{P}(C(X') \in B_i) \pm \delta$, or equivalently, $p_i^+ = e^{\pm\varepsilon} \cdot p_i^- \pm \delta$. Since $\delta \leq \varepsilon \leq 0.1$, this means $|p_i^+ - p_i^-| \leq 0.3$. Also, since $p_i = \frac{1}{2}(p_i^- + p_i^+)$, this means $p_i^+ - p_i \leq 0.15$.

Now, since the points in $X_0$ are separated by $\frac{\rho}{10}$, the balls $B_i$ are disjoint. So, for any fixed $C$, at most $k$ of the events of some point in $C$ is in $B_i$ can hold. Therefore, $\sum_{i=1}^{2k} p_i \leq k$, which means $\sum_{i=1}^{2k} p_i^+ \leq k + 0.15 \cdot 2k = 1.3k$.

Now, $\text{cost}(X, C)$ is at least $\sum_{i=1}^{2k} \left(\frac{\rho}{10}\right)^p \cdot \mathbb{P}(x_i \in X) \cdot (1 - p_i^+)$. This is because $\mathbb{P}(x_i \in X) \cdot (1 - p_i^+)$ represents the probability that $x_i \in X$ but no point in $C$ is within $\frac{\rho}{10}$, so the point $x_i$ itself contributes $\left(\frac{\rho}{10}\right)^p$ to the cost. But this simply equals $\left(\frac{\rho}{10}\right)^p \cdot (2k - \sum_{i=1}^{2k} p_i^+) \geq \Omega(\rho^p \cdot k)$, as desired. $\square$

## E  Additional Details of Experiments

### E.1  Details of Implementations

**More Implementation Details.**  Note that the privacy budget is consumed in 3 parts: (1) computing $\bar{X}$, (2) computing $count(g)$ for cells $g$ at level $l$ for $l \in [L_1 + 1, L_2]$, and (3) computing DP semi-coreset $\tilde{X}_j$ in Algorithm 1. We split the privacy budget uniformly, i.e., each part takes $\varepsilon/3$ and $\delta/3$.

**Detailed implementation of Algorithm 2.**  Since we know each $x$ is in $[-1, 1]^d$, When we compute $\tilde{x}_i$, if any coordinate is outside $[-2, 2]$, we project it to $[-2, 2]$. We choose $REP = 5$. The random shifted vector is chosen uniformly random from $[0, 4]^d$ and thus the cell in the highest level of the

quadtree has side length 8. We choose $L_1 = 5$ and $L_2 = 10$. When we compute $count(g)$ of cell $g$ at level $l \in [L_1 + 1, L2]$, we apply Gaussian thresholding mechanism[5].

**Detailed implementation of Algorithm 1.** We set $S = \sqrt{2 \log(1.25)/(\delta/6)} \cdot \sqrt{d}/(\varepsilon/6)$. We run the first loop of Algorithm 1 to obtain $\tilde{X}_0$. We slightly modify the second loop as follows: $\hat{X}_j = \{x_i \in X \setminus \tilde{X}_0 \mid d(x_i, f_j) = d(x_i, F)\}$. We use Gaussian thresholding mechanism to compte $\hat{n}_j$ to estimate $|\hat{X}_j|$. If $\hat{n}_j \leq 0$, we drop $\hat{X}_j$. Otherwise we run a semi-coreset for $\hat{X}_j$. It is easy to show that the above modifed procedure is still DP. When we use the DP open-source library [6] to compute the (semi)-coreset of $\hat{X}_j$, we specify the the bounding ball is centered at $f_j$ with radius $\min(S, \sqrt{(d)}) \cdot \rho$, i.e., the points in $\hat{X}_j$ that are outside the ball are projected to the ball.

Finally, we use non-DP baseline $k$-means to run $k$-means over the union of (semi)-coresets $\tilde{X}_1 \cup \cdots \cup \tilde{X}_{\alpha \cdot k}$ and $\tilde{X}_0$.

### E.2  Preprocessing Steps of the Datasets

Dataset gowalla contains 6,442,890 user check-ins of 107,092 different users and dataset brightkite contains 4,491,143 user check-ins of 51,406 different users. Each check-in record contains a location information (latitude and longitude). For each user, we use its latest check-in record, and thus we obtain a dataset of size 107,092 x 2 for gowalla and a dataset of size 51,406 x 2 for brightkite. For each latitude, we divided it by 90. For each longitude, we divided it by 180. Thus, each coordinate of a user is in [-1,1].

For other non-geographic datasets (shuttle, skin, rangequeries, s-sets), we follow the same preprocessing steps of experiments in [24]. In particular, we linearly rescale each dimension of each point to make the coordinate have value in [-1,1].

## F  Broader Impacts

Our work developed distance based private algorithms for clustering problems. Distance based privacy provides provable standards of privacy but its use, like that of any privacy protection, is subject to limitations (we refer to standard textbooks on differential privacy such as [32] for the subject). We also stress that privacy is only one of the requirements of a responsible machine learning system. For this reason, we encourage anyone using the techniques developed in this paper in a real system, to review carefully the overall safety of their design.

---

[5]`https://github.com/google/differential-privacy/blob/main/common_docs/Delta_For_Thresholding.pdf`.

[6]`https://ai.googleblog.com/2021/10/practical-differentially-private.html`.

