# OpenReview forum: "$k$-Means Clustering with Distance-Based Privacy"
_NeurIPS.cc/2023/Conference — NeurIPS 2023 poster_

### Official Review · Reviewer_CEv9 · 2023-07-01

**Soundness:** 3 good
**Presentation:** 1 poor
**Contribution:** 2 fair
**Rating:** 5
**Confidence:** 3

**Summary:**

The present paper focuses on the application of a distance-based privacy notion called rho-dist-DP in the context of clustering. This privacy notion aims to protect an individual data point that moves at most a specified distance (rho) in the underlying metric space. The authors demonstrate that by leveraging rho-dist-DP, it is possible to achieve better utility compared to traditional DP approaches. To this end, they propose a new algorithm and provide both theoretical and empirical evidence supporting its improved performance over standard DP clustering algorithms. At a technical level, the main theoretical result is a generalization of [70] whose proof is inspired by techniques in [22,25].

**Strengths:**

The notion of distance-based DP presents a promising alternative to address the high utility cost associated with traditional DP. Effectively designing and analyzing distance-based DP algorithms is crucial for implementing privacy in applications that demand a high level of utility. The present paper contributes towards advancing research in this direction.

**Weaknesses:**

The primary inquiry addressed in this paper revolves around whether a better performance can be achieved by relaxing DP into a distance-based notion. Regrettably, the answer to this question appears to be rather straightforward: yes. The reason behind this outcome is rather evident—by setting rho to be the diameter of the underlying metric space (Lambda), regular DP can be recovered. Furthermore, the lower bound in Appendix D shows that rho-dist-DP is in fact DP in this specific case. As a result, the driving question of this paper and its subsequent answer feel rather dull.

Additionally, a significant aspect that the present paper overlooks is the interpolating nature of the distance parameter, rho. Specifically, when rho equals 0, no privacy is provided, while setting rho to Lambda recovers DP. Although Theorem 1.1 suggests a monotonic relationship between performance and the distance parameter rho, the experimental results fail to consistently support this claim. In fact, the distance-based DP algorithm (with rho < Lambda) is at times outperformed by the DP algorithm (with rho = Lambda), thus contradicting the expected trend. (In a slightly related note, it may be more insightful to compare rho-dist-DP against d_X-privacy [41] rather than classical DP, as the latter serves only as a limiting case for rho-dist-DP.)

From a writing perspective, Sections 3-5 exhibit shortcomings in terms of explaining the proposed algorithm. The absence of illustrations or concrete examples hampers clarity and understanding. Instead, the authors provide an abstract explanation that does more harm than good. While it is reasonable to assume that a meaningful message can be reconstructed from the current discussion, it is crucial for the authors to provide it directly to the reader in a more accessible manner.

**Questions:**

In its present form, the paper fails to capture the interpolating nature of the distance parameter (rho). Given its importance, it would be helpful to have solid theoretical results and strong empirical evidence on the role of this parameter.

**Limitations:**

The authors do not address the limitations of the proposed method carefully. As mentioned in the Weaknesses Section, the interpolation character of the distance parameter (rho) is not carefully addressed, which is fundamental to understand the consistency of the proposed algorithm.

---

> ### Author Rebuttal · Authors · 2023-08-09
>
> We thank the reviewers for the valuable comments.
>
> Q: “The primary inquiry addressed in this paper revolves around whether a better performance can be achieved by relaxing DP into a distance-based notion. Regrettably, the answer to this question appears to be rather straightforward: yes. The reason behind this outcome is rather evident—by setting rho to be the diameter of the underlying metric space (Lambda), regular DP can be recovered. Furthermore, the lower bound in Appendix D shows that rho-dist-DP is in fact DP in this specific case. As a result, the driving question of this paper and its subsequent answer feel rather dull.”
>
> A: Any relaxation of a constraint, implies that a not-worse performance is possible. This is not surprising. What is interesting, is to show how the improved performance can be obtained (our algorithm) and by how much such a method outperforms the non-relaxed version. This is common throughout the theory of optimization and in DP as well.
>
> Let’s consider an historic example: Approximate DP, aka (epsilon,delta)-DP. This is obviously a relaxation of Pure DP (which is obtained by setting delta=0) and was introduced in the landmark Dwork et al 2006 “Our data, ourselves: Privacy via distributed noise generation”. Obviously any problem admits a better solution with (epsilon,delta)-DP than in epsilon-DP. That does not diminish the interest in studying it, because the non-trivial question is how to obtain such improved performances and when such improved performances are possible. The study of the (epsilon,delta)-DP relaxation introduced significant innovation to the field of DP (e.g., the Guassian mechanism which is not possible in epsilon-DP).
> We believe that studying rho-dist-DP can contribute to novel results in metric space data beyond our work.
>
>
> Q: Additionally, a significant aspect that the present paper overlooks is the interpolating nature of the distance parameter, rho. Specifically, when rho equals 0, no privacy is provided, while setting rho to Lambda recovers DP. Although Theorem 1.1 suggests a monotonic relationship between performance and the distance parameter rho, the experimental results fail to consistently support this claim. In fact, the distance-based DP algorithm (with rho < Lambda) is at times outperformed by the DP algorithm (with rho = Lambda), thus contradicting the expected trend.
>
> A: We do not think we overlooked the interpolating nature of the distance parameter rho. In fact both our theoretical result (Theorem 1.1) and our empirical result (Figure 2, and we apologize about the typo in Figure 2 where "dx clustering" should be "dist-DP k-means") supports the interpolating nature of the distance parameter, rho. In particular when rho=0, we exactly recover the result as non-private k-means++ (as shown in Figure 2). In addition, as shown in Figure 2, when rho decreases, we get better empirical approximation guarantee. In Figure 1, we are in the case that rho < Lambda and our algorithm (yellow line) has better empirical approximation than the previous DP algorithm (green line).
>
> When rho = Lambda, our algorithm might be outperformed by the previous DP algorithm. This is because our algorithm needs to utilize the information of the parameter rho, and it splits the privacy budget to handle points “far” from the centers and the points “close” to the centers separately, which introduces some additional overheads in the noise. To develop an (eps,delta,rho)-dist DP clustering algorithm that has exactly the same empirical approximation as the (eps,delta)-DP clustering algorithm when rho=Lambda is an interesting open question, and we leave it as a future work. In general the regime of interest for our algorithm are the cases where \rho << \Lambda as otherwise the two definitions converge and it is possible to use off-the-shelf DP clustering algorithm.
>
>
>
> Q: In a slightly related note, it may be more insightful to compare rho-dist-DP against d_X-privacy [41] rather than classical DP, as the latter serves only as a limiting case for rho-dist-DP.
>
> A: As we discussed before, though a DP algorithm implies a rho-dist-DP algorithm, the core technical question of our paper is how to develop an algorithm with better approximation guarantees in the rho-dist-DP setting (a more relaxed setting) than the algorithms in the standard DP setting (a more restrictive setting). We observe that dX is an orthogonal relaxation of DP, so it is not clear for us to compare rho-dist-DP setting with dX-privacy setting since dX-privacy depends on the definition of the distance between two datasets, where our definition depends on the distance between the different data points in two neighboring datasets.

---

> > ### Author Response · Authors · 2023-08-14
> > **Reviewer CEv9 -- Any comment on the rebuttal?**
> >
> > Dear Reviewer CEv9,
> > As the deadline for the interactive phase draws to a close we would like to ask your feedback on the rebuttal. Have we addressed your concerns or do you have any additional questions?
> > Best regards

---

> > > ### Comment · Reviewer_CEv9 · 2023-08-15
> > >
> > > Thank you for your detailed response. While your reply clarified some points, the paper still gives a sense of preliminarity regarding the "interpolating nature" of rho. As you acknowledged, when rho = Lambda, your algorithm might be outperformed by the previous DP algorithm. I slightly raised my score and will take your comments into consideration during the discussion.

---

### Official Review · Reviewer_1C7X · 2023-07-05

**Soundness:** 2 fair
**Presentation:** 2 fair
**Contribution:** 2 fair
**Rating:** 4
**Confidence:** 4

**Summary:**

This paper proposes a definition called "distance-based privacy" which is relaxation of differential privacy. Their definition differs from standard differential privacy in its notion of neighboring instances; whereas standard DP allows an arbitrary replacement of a single item from the space, their definition considers as neighbors only those instances that can be obtained by changing an item up to a Euclidean distance of $\rho$ from the original. They then study k-means/median clustering under this relaxed definition and propose an algorithm for the problem built upon previous work. They also provide analysis and experimental results for their algorithm.

**Strengths:**

- This paper is generally well-structured.
- They provide experimental results on several real-world datasets

**Weaknesses:**

- The relaxed definition is a strictly local definition, in that it only provides privacy protection up a pre-specified distance. This in itself is not an issue. However, by the way it's defined, this means for cities B and C, both within a distance $\rho$ from city A, an algorithm which allows indistinguishability between B and A, does not provide protection for locations in C. Moreover, if dist(B,A) > dist(C,A), a different $\rho$ would be required if indistinguishability of A from B is desired than if that from C is desired (somewhat reminiscent of local sensitivity..). Either that, or larger $\rho$ needs to be used, which eventually becomes the radius of the space. Also, essentially the same definition was introduced in [a] for privately releasing locations.
- When comparing with an algorithm (e.g. DP k-means in the paper) which satisfies the stronger notion of standard DP, it should be discussed whether the competitor algorithm can also use the relaxed notion of neighbors to its advantage, or otherwise discuss whether the privacy parameters should be adjusted in the experiments for a fair comparison.
- Since the proposed algorithm is built upon a previous algorithm [25], the experiments should also include performance of [25] to show the proposed modification is indeed an improvement (or otherwise discuss why it's not possible to do so).
- Parts of the writing appear to be re-phrasing of that from [25] (e.g. part of the introduction). Moreover, some sentences appear to be taken verbatim from [25] (e.g. lines 23, 140-142, 146-150, 155-164...). This reduces the credibility of the work.

[a] Lu Zhou et al. "Achieving Differentially Private Location Privacy in Edge-Assistant Connected Vehicles."

**Questions:**

N/A

---

> ### Author Rebuttal · Authors · 2023-08-09
>
> We thank the reviewers for the valuable comments.
>
> Q: The relaxed definition is a strictly local definition [...] However, by the way it's defined, this means for cities B and C, both within a distance …
>
> A: We would like to recap the definition and the properties of (eps, delta, rho)-dist-DP to address some misunderstandings. Note that in our definition of (eps, delta, rho)-dist-DP, if we move an *arbitrary* data point by a distance at most rho, the output distribution should always be (eps, delta)-close to the original output distribution. This means that if d(A,B) < rho, it is hard to distinguish that a user is from city A or from city B, and similarly, if d(A,C) < rho, it is hard to distinguish that a user is from city A or from city C as well (which means that an (eps,delta,rho)-dist DP algorithm protects all location information up to distance rho at the same time for *all* users). In addition, d(A,B) < rho and d(A,C) < rho this trivially implies that d(B, C) < 2*rho by triangle inequality. Therefore, the output distribution when a user is at B is be (2*eps, 2*delta)-close to the output distribution when the user is moved to C. In this example, even if B and C are farther than \rho there is a privacy protection although with slightly weaker guarantees.
>
> In general, the definition automatically implies (L*eps, L*delta)-DP if you move a single point a distance L*rho. Of course if a point is moved between two very far location >>\rho (e.g,  for \rho = 1 mile, and a point moves from the East coast of the US to the West Coast of the US) the privacy protection degrades to the point that it possible to guess with higher likelihood to which side of the country the point belongs. This is expected as this is privacy relaxation and in some contexts for appropriate \rho this privacy protection is sufficient.
> Notice that variants of DP in a similar fashion (e.g., Dx privacy) have been used already in the past on location data (see cited work) and the same phenomena applies to them.
> What is an appropriate \rho for a given application is policy question beyond our work. Similarly to process is in place for determining eps and delta in any real DP application (see for instance the complex discussion on the epsilon parameter in 2020 Census, https://www.census.gov/newsroom/press-releases/2021/2020-census-key-parameters.html) this involves assessment of privacy risk, utility of the system, as well as the legal, policy and regulatory environment. Such policy discussions are outside of the scope of the paper (See the review from vkqo06 that agrees with this point).
>
>
> Q:  When comparing with an algorithm (e.g. DP k-means in the paper) which satisfies the stronger notion of standard DP, it should be discussed whether the competitor algorithm can also use the relaxed notion of neighbors to its advantage, or otherwise discuss whether the privacy parameters should be adjusted in the experiments for a fair comparison.
>
> A: The main message of our paper is that our algorithm utilizes the relaxed definition of the neighboring dataset to achieve a better approximation from both theoretical and empirical perspectives, and none of the previous algorithms can utilize such relaxation. Keeping eps and delta the same for both our algorithm and the competitor algorithm is fair, since this keeps the same tolerance for the difference in the output distributions for neighboring datasets. We will add this clarification in our empirical study section in the final version of our paper.
>
> Q: Since the proposed algorithm is built upon a previous algorithm [25], the experiments should also include performance of [25] to show the proposed modification is indeed an improvement (or otherwise discuss why it's not possible to do so).
>
> A: Note that though some of our analysis techniques are inspired by [25], we have very different algorithmic structures. In addition, [25] mainly focuses on distributed/parallel settings which introduces complexity that we do not have and would make the comparison not fair. Note that [25] provides a general framework which uses any non-distributed/non-parallel DP k-means/k-median algorithm as a black box subroutine, and their approximation guarantee is worse than the black box algorithm they used. This is why we only need to compare our algorithm with non-distributed/non-parallel DP k-means/k-median algorithms.
> Furthermore, we want to emphasize that the work of [25], has not been implemented. The paper [25] is purely theoretical and does not have an empirical validation section. Implementing it in all its details is highly non-trivial given its complexity.
>
> Q: Parts of the writing appear to be re-phrasing of that from [25] (e.g. part of the introduction). Moreover, some sentences appear to be taken verbatim from [25] (e.g. lines 23, 140-142, 146-150, 155-164...). This reduces the credibility of the work.
>
> A: We apologize for the sentences that appear too close to the work of [25]. The introduction shares commonality to that work which inspired ours as we point out several times. We will thoroughly improve our presentations.

---

> > ### Author Response · Authors · 2023-08-14
> > **Reviewer 1C7X -- Any comment on the rebuttal?**
> >
> > Dear Reviewer 1C7X,
> > As the deadline for the interactive phase draws to a close we would like to ask your feedback on the rebuttal. Have we addressed your concerns or do you have any additional questions?
> > Best regards

---

> > ### Comment · Reviewer_1C7X · 2023-08-16
> >
> > Thank you for your response.
> >
> > About the experiments, I agree that the effectiveness of utilizing $\rho$ should be demonstrated, but that can be shown by varying $\rho$ from 1 to a much smaller value, such as shown in Fig. 2. In order to better understand the trade-off in switching from DP to $\rho$-dist DP, I think it helps to advantage the DP competitor algorithm with a larger privacy budget (say $2\varepsilon$, $3\varepsilon$). For example, if the DP competitor algorithm can already match the performance of the $\rho$-dist DP algorithm with $2\varepsilon$ (say), then there is no compelling reason to switch to an algorithm with a weaker privacy guarantee?

---

> > > ### Author Response · Authors · 2023-08-16
> > >
> > > We would like to clarify the relationship between dist-DP and DP a bit more to address a misunderstanding. As we mentioned in the rebuttal, suppose the diameter Lambda = L * rho, then an (eps,delta,rho)-dist-DP algorithm implies an (L*eps, L*delta)-DP algorithm. However, the reverse is not true, i.e., if an algorithm is (L*eps, L*delta)-DP, it is not necessarily an (eps,delta,rho)-dist-DP algorithm. Therefore, when comparing (eps,delta,rho)-dist-DP with (L*eps, L*delta)-DP, our dist-DP algorithm provides a stronger privacy guarantee instead of a weaker privacy guarantee.
> > >
> > > To give a concrete example, suppose the dataset contains geographic locations of users on the earth, then the diameter (the furthest distance between two points on the earth) of the dataset is ~20k kilometers. The distance between New York and Toronto is ~500 kilometers. It means that if we move one user from New York to Toronto, then the probability density of every possible output of a (500 kilometer)-dist-DP algorithm with eps=0.1, will shift by a multiplicative factor at most e^{0.1} (which is roughly 1.1). However, when we consider an (0.1 * 40)-DP algorithm, even if we only move the location of a user by distance at most 1 meter, it is possible that the probability density of possible outputs is shifted by a multiplicative factor > 50 and thus the attacker may more easily obtain information about the user location up to 1 meter of distance.
> > >
> > > Intuitively, the promise of our algorithm with small epsilon is that we strongly protect knowing the precise location of a point up to <=\rho precision. Setting \rho=500km and small epsilon will protect knowledge of which city the user is in with very strong bounds (but will not protect which hemisphere the user is in with meaningful bounds). In comparison, a large epsilon-DP algorithm will not protect any disclosure on the user with meaningful bounds (up to learning the precise GPS location of the user).
> > >
> > > Please let us know if you have any additional questions or concerns.

---

> > > > ### Comment · Reviewer_1C7X · 2023-08-17
> > > >
> > > > The part about $(\varepsilon,\delta,\rho)$-dist DP implying $(L\varepsilon,L\delta)$-DP is fine. However, your experiments compare $(\varepsilon,\delta,\rho)$-dist DP with $(\varepsilon,\delta)$-DP, right? And the latter is a stronger privacy guarantee because it holds for $\rho=\Lambda$?

---

> > > > > ### Author Response · Authors · 2023-08-17
> > > > >
> > > > > Yes, (eps,delta)-DP is a stronger privacy guarantee than (eps,delta,rho)-dist DP. But (eps,delta,rho)-dist DP is a stronger privacy guarantee than (L*eps,L*delta)-DP if Lambda = L * rho.
> > > > >
> > > > > We compare our (eps,delta,rho)-dist DP algorithm with (eps,delta)-DP algorithm because we want to show that when preserving the location information of a user up to distance rho is acceptable, we are able to get better approximation guarantee.
> > > > >
> > > > > Note that as we discussed in the rebuttal, even though (eps,delta,rho)-dist DP is weaker than (eps,delta)-DP, to develop an algorithm utilizing the distance threshold rho to achieve a better approximation guarantee is still an interesting question.

---

> > > > > > ### Author Response · Authors · 2023-08-21
> > > > > >
> > > > > > Dear reviewer 1C7X,
> > > > > > We would like to ask if we have fully addressed your doubts and if otherwise you have any questions.
> > > > > > Best regards

---

### Official Review · Reviewer_vkqo · 2023-07-06

**Soundness:** 3 good
**Presentation:** 3 good
**Contribution:** 3 good
**Rating:** 6
**Confidence:** 4

**Summary:**

In this paper, the author proposed efficient (ε, δ, ρ)-dist-DP algorithms for k-means and k-median problems to protect the privacy of exact locations as well as achieving good performance.

**Strengths:**

The author proposes new efficient (ε, δ, ρ)-dist-DP algorithms of k-means and k-median problems, which successfully protect the privacy of exact locations as well as achieving good performance. The author explains the proposed method clearly and detailly.

**Weaknesses:**

The structure of this paper needs to be carefully reconsidered. For example, the “Running time and parallel computation” in introduction should be put in latter experiment section. The motivation and contributions of this paper is not clearly written. There is a lack of overall conclusion at the end of this paper.

**Questions:**

1. What is PDF in line 141? The definition of abbreviation should be added before using it.
2. Why does the privacy parameters set to be ε = 1, δ = 10-6 ?
3.At the beginning of Section 3, the author should give a brief summary of this section before introducing the content in following sections.

**Limitations:**

As the author argues, the choices of ε, δ and ρ is difficult and remain open. These privacy parameters should not be determined by our algorithm, but rather by legal teams, policy makers, or other experts for different specific scenarios. This is an expert determination that is outside of the scope of this paper but has been studied by practitioners extensively.

---

> ### Author Rebuttal · Authors · 2023-08-09
>
> We thank the reviewers for the valuable comments.  We will apply the suggestions on the structure of the paper to improve readability.
>
> Q: What is PDF in line 141? The definition of abbreviation should be added before using it.
>
> A: PDF means probability density function - we will update the paper to clarify this.
>
> Q: Why does the privacy parameters set to be ε = 1, δ = 10-6 ?
>
> A: We experiment with these parameters as they are standard settings used in most DP papers. Eps around 1 is a standard benchmark, while delta is often set to be around 10-6 (or in the order of 1 over the size of the data).

---

> > ### Author Response · Authors · 2023-08-14
> > **Any comment on the rebuttal?**
> >
> > Dear Reviewer vkqo,
> > As the deadline for the interactive phase draws to a close we would like to ask your feedback on the rebuttal. Is there any  additional questions?
> > Best regards

---

> > > ### Author Response · Authors · 2023-08-21
> > >
> > > Dear reviewer vkqo,
> > > We would like to ask before the deadline, if we have fully addressed your questions.
> > > Best regards

---

### Official Review · Reviewer_nn8B · 2023-07-06

**Soundness:** 3 good
**Presentation:** 3 good
**Contribution:** 3 good
**Rating:** 8
**Confidence:** 4

**Summary:**

The paper studies the problems of solving k-means and k-median under a restricted location
privacy model, in which privacy is protected when a point moves by at most distance \rho.
It gives algorithms for these clustering problems with additive errors which are a function
of \rho, instead of the diameter \Lambda, improving on the usual DP results for these problems.
The authors also give a lower bound that depends on \rho. The algorithm shows improvement
over the standard DP algorithm in experiments.

**Strengths:**

The proposed method gives a significant improvement in performance over the prior algorithms
for k-median and k-means with privacy, with additive error being a function of \rho instead
of \Lambda. This can be quite significant in some settings, where this notion of privacy
is reasonable. The authors give rigorous privacy and accuracy guarantees, and the technical ideas
are quite interesting. The experimental results show some interesting behavior.
The presentation is generally quite nice.

**Weaknesses:**

It is not clear how \rho would be chosen for this model to be used. The authors do not consider
this point either in the theoretical part, or through the experiments. The paper builds heavily on
ideas from a prior work, though there are some new aspects.

**Questions:**

lines 35-36: it might be useful to put citations for the results of this type, e.g., [47]

line 56: here it is mentioned that a point moves distance \rho in a mtric space, but the whole
paper is on Euclidean space. It would be useful to explain this further, or make this restricted.

The related work section is quite limited, and could do with some deeper discussion about the
related work, including technical aspects of the papers cited here and in the earlier sections
of the paper. It might also be useful to discuss other notions which have been proposed to deal
with high sensitivity, e.g., Lipshitz extensions (Raskhodnikova and  Smith, 2015), which have
been used in the context of graph statistics and other problems.

Is there an extension of the notion of \rho-dist-DP for more general metric spaces (instead
of Euclidean), and similar upper and lower bounds? One might need to put constraints on distance
with respect to other points as well.

line 148: it might be useful to note as in [43] that TLap has 0 probability outside this interval,
and it is a valid probability distribution

line 149: "is known" ("it" is missing)

paragraph in lines 192-198: if only a bounded number of heavy cells from each level are added,
why would the error depend on \Gamma?

line 227: "the noisy version" ("is" missing)

line 339: "partitioning straties"

line 341: "investage"

How do the results depend on \epsilon?

In Figure 1, there is a gap between the dist-DP and DP k-means, but it is not that big. Is that to
be expected, based on these settings and datasets?

In Figure 2, all the plots seem to have a significant increase at 0.08. Is this kind of
behavior expected? This might help in deciding the value of \rho, which is a parameter right now


**Limitations:**

To some extent. There are no negative social impacts

---

> ### Author Rebuttal · Authors · 2023-08-09
>
> We thank the reviewers for the valuable comments.
>
> Q: It is not clear how \rho would be chosen for this model to be used. The authors do not consider this point either in the theoretical part, or through the experiments.
>
> A: As we discussed in Section 7, rho is a free privacy parameter like epsilon and delta in DP. As such it should be determined by those in charge of the privacy policy of the real-world system implementing an algorithm like ours. A similarly process is in place for determining eps and delta in any real DP application (see for instance the complex discussion on the epsilon parameter in 2020 Census, https://www.census.gov/newsroom/press-releases/2021/2020-census-key-parameters.html). In the real-world, these decisions are based on a complex assessment of privacy risk, utility of the system, as well as the legal, policy and regulatory environment. In this work, we focus on the algorithm aspects of the problem and such policy discussions are outside of the scope of the paper (See the review from vkqo06 that agrees with this point). Our goal is to show that it is possible to get a better approximation guarantee for the notion of distance-based privacy and that our algorithm can impose any \rho guarantee desired by the decision maker. For the improved results we provide both theoretical and empirical justification.
>
> Q: Regarding other related work.
>
> A: In this work, we mainly focus on the definition of a variant of differential privacy. The study of how to deal with high sensitivity instances under standard differential privacy is orthogonal to our work. But we agree that discussion of these lines of work would give readers a more complete picture of the literature and thus we will include the discussion of these related works in our final version of the paper.
>
> Q: Regarding Distance-based privacy for general metric space
>
> A: Thank you for pointing it out. The notion of distance-based privacy can be over any metric space. But in our paper, we only consider Euclidean space. We will clarify this in the final version of our paper. The lower bound should apply as well (since the lower bound for Euclidean space also applies for general metric space).
> For the upper bound, since our algorithm itself heavily relies on the properties of the Euclidean space, it may not be directly applicable to general metric space. We leave the clustering problem in general metric space under distance-based privacy as a future work.
>
> Q: Doubts on Line 192-198.
>
> A: We assume that you are asking “why does it depend on \Lambda?”. Consider a dataset with a few outlier points, and consider the level where each cell has side length O(\sqrt{d}*\Lamda). Then there should be O(k) non-empty cells, and in the non-DP setting, we are able to find all of them. However, after we add noise to the count of points in each cell, we may miss the cell which contains the outlier due to the noise, which may cause an Omega(\Lambda) additive error.
>
> Q: How does the result depend on eps?
>
> A: The dependence is polynomial in epsilon, specifically proportional to 1/eps^{3.01} with our parameter settings. The constant 3.01 be replaced by 3+\eta for any arbitrarily small constant \eta. We will update the paper to clarify the dependence.
>
>
> Q: In Figure 1, there is a gap between the dist-DP and DP k-means, but it is not that big. Is that to be expected, based on these settings and datasets?
> A:  It is as expected given rho and datasets. Note that we only show one fixed rho when comparing with DP k-means, and the point is to show that our clustering cost is already better than DP clustering cost when rho is small enough. If we choose rho to be much smaller, (as presented in Figure 2), our clustering cost can be much smaller (i.e., close to the non-private k-means) and thus can be much better than DP k-means.
>
> Q: In Figure 2, all the plots seem to have a significant increase at 0.08. Is this kind of behavior expected? This might help in deciding the value of \rho, which is a parameter right now
>
> A: Our conjecture is that it depends on the structure of the dataset itself. For deciding \rho please refer to the discussion above. Utility of the system certainly (appropriately obtained) can be an input in the decision process but must be considered in relation to the risk factors and other considerations.
>
> We will address other comments of writing and presentation in the final version of the paper.

---

> > ### Comment · Reviewer_nn8B · 2023-08-19
> >
> > Thanks to the authors for the detailed responses.

---

> ### Author Response · Authors · 2023-08-14
> **Reviewer nn8B -- Any comment on the rebuttal?**
>
> Dear Reviewer nn8B,
> As the deadline for the interactive phase draws to a close we would like to ask your feedback on the rebuttal. Is there any  additional questions?
> Best regards

---

### Decision · Program_Chairs · 2023-09-21

**Decision:**

Accept (poster)

**Comment:**

This interesting paper shows how to adapt $k$-means clustering to privacy considerations. I urge the authors to use the camera ready time and extra page to carefully adjust the paper to reflect the extensive reviewer discussions.